# NINJ1 induces plasma membrane rupture and release of damage-associated molecular pattern molecules during ferroptosis

Saray Ramos (ID), Ella Hartenian (ID), José Carlos Santos (ID), Philipp Walch (ID) & Petr Broz (ID) ✉

## Abstract

**Ferroptosis is a regulated form of necrotic cell death caused by iron-dependent accumulation of oxidized phospholipids in cellular membranes, culminating in plasma membrane rupture (PMR) and cell lysis. PMR is also a hallmark of other types of programmed necrosis, such as pyroptosis and necroptosis, where it is initiated by dedicated pore-forming cell death-executing factors. However, whether ferroptosis-associated PMR is also actively executed by proteins or driven by osmotic pressure remains unknown. Here, we investigate a potential ferroptosis role of ninjurin-1 (NINJ1), a recently identified executor of pyroptosis-associated PMR. We report that NINJ1 oligomerizes during ferroptosis, and that *Ninj1*-deficiency protects macrophages and fibroblasts from ferroptosis-associated PMR. Mechanistically, we find that NINJ1 is dispensable for the initial steps of ferroptosis, such as lipid peroxidation, channel-mediated calcium influx, and cell swelling. In contrast, NINJ1 is required for early loss of plasma membrane integrity, which precedes complete PMR. Furthermore, NINJ1 mediates the release of cytosolic proteins and danger-associated molecular pattern (DAMP) molecules from ferroptotic cells, suggesting that targeting NINJ1 could be a therapeutic option to reduce ferroptosis-associated inflammation.**

**Keywords** Ferroptosis; Cell Death; Plasma Membrane Rupture; Ninjurin-1; Inflammation
**Subject Categories** Autophagy & Cell Death; Membranes & Trafficking

## Introduction

Programmed or regulated cell death is important for organismal development, the maintenance of tissue homeostasis, and the removal of infected or malignant cells (Green, 2011). Over the last decades, numerous types of programmed cell death have been identified, ranging from immunologically silent apoptosis to highly inflammatory forms of programmed necrosis, such as pyroptosis, necroptosis and NETosis (Galluzzi et al, 2018). Ferroptosis is an emerging form of regulated necrosis that is characterized by an iron-dependent formation of lipid peroxides in cellular membranes (Dixon et al, 2012; Jiang et al, 2021). Under physiological conditions, the build-up of such lipid peroxides is counteracted by antioxidant enzymes that prevent lipid peroxidation or reduce oxidized lipids in membranes to lipid alcohols. Among such detoxification mechanisms are GPX4 (glutathione peroxidase 4), which uses glutathione as a co-factor to reduce peroxidized lipids, and FSP-1 (Ferroptosis Suppressor Protein 1) (Doll et al, 2019; Bersuker et al, 2019), which catalyzes the reduction of coenzyme Q10 that in turn can reduce lipid peroxides. Ferroptosis can be induced by triggers that either inhibit these pathways (e.g., RAS-selective lethal 3 (RSL3), ML162 and Erastin (Dixon et al, 2012)), or overwhelm them by inducing uncontrolled lipid peroxidation (e.g., Tert-Butyl hydroperoxide (TBOOH) or Cumene hydroperoxide (CuOOH) (Wenz et al, 2018)). Ferroptosis has been shown to contribute to many human pathologies including ischemia/reperfusion injury, neurodegenerative diseases, cancer and inflammatory disorders (Stockwell et al, 2017). A growing body of evidence suggests that ferroptosis involvement in disease is linked to the inflammation caused when ferroptotic cells release lipid peroxidation products and cytosolic proteins that act as damage-associated molecular patterns (DAMPs) (Sun et al, 2020; Sarhan et al, 2018; Proneth and Conrad, 2019).

Morphologically, ferroptotic cells display signs of membrane blebbing and ballooning (Dixon et al, 2012; Stockwell et al, 2017), similar to cells undergoing necroptosis and pyroptosis. The latter forms of programmed necrosis involve the activation of specific cell death executors, such as mixed-lineage kinase like (MLKL) (Sun et al, 2012; Murphy et al, 2013; Wang et al, 2014) and gasdermin D (GSDMD) (Kayagaki et al, 2015; Shi et al, 2015), respectively, which form plasma membrane pores to initiate cellular demise. By contrast, it is not yet known if ferroptosis involves the activation of a pore-forming protein, or if membrane damage during ferroptosis is solely caused by peroxidized lipids that alter the physical properties of cellular membranes, thus rendering them less stable. Interestingly, two recent studies showed that lipid peroxidation is the earliest observable event during ferroptosis and that it is rapidly followed by calcium influx (Pedrera et al, 2021; Riegman et al, 2020). Importantly, calcium influx was found to precede a progressive loss of membrane integrity (measured by the uptake of small DNA-binding dyes such propidium iodide (PI)), and

Department of Immunobiology, University of Lausanne, Epalinges, Switzerland. ✉E-mail: petr.broz@unil.ch

complete cell lysis (measured by Lactate dehydrogenase (LDH) release). If the formation of small membrane lipid lesions or an opening of ion channels caused the calcium influx was not addressed by these studies. Follow-up work showed that ion fluxes during ferroptosis are most likely caused by the opening of membrane channels such as the piezo-type mechanosensitive ion channel component 1 (Piezo-1) channel, which responds to increased membrane tension due to lipid peroxidation, or transient receptor potential (TRP) channels (Hirata et al, 2023). Interestingly, it was also reported that large osmoprotectants can block calcium influx and subsequent events, giving rise to the hypothesis that ferroptosis involves the formation of a yet elusive 'ferroptotic pore' that drives water influx and osmotic cell lysis (Pedrera et al, 2021; Riegman et al, 2020). However, whether the ferroptotic pore corresponds to lesions caused by peroxidized lipids, or if it is formed by a protein remains unknown.

Plasma membrane rupture (PMR) is a defining and terminal event in many forms of cell death (Zhang et al, 2018; Hiller and Broz, 2021). While it was long thought to occur passively as a result of rising osmotic pressure, it recently emerged that PMR is actively executed by the plasma membrane protein ninjurin-1 (NINJ1) (Kayagaki et al, 2021). During pyroptosis, NINJ1 is activated downstream of GSDMD pore formation and oligomerizes into filamentous assemblies (Kayagaki et al, 2021; Degen et al, 2023). These NINJ1 filaments have a distinct hydrophilic and hydrophobic face, allowing them to cap membrane edges and disrupt membranes, thus forming lesions. NINJ1 also mediates PMR in cells treated with pore-forming toxins, in late-stage apoptotic cells (e.g., secondary necrosis) and even partially contributes to the lysis of necroptotic cells (Kayagaki et al, 2021). However, it has not yet been shown if NINJ1 is activated during ferroptosis and if it contributes to the lysis of ferroptotic cells.

Here we investigated the role of NINJ1 in ferroptosis upon treatment with CuOOH, RSL3 or ML162 in human and mouse cells. We found that upon induction of ferroptosis, plasma membrane NINJ1 rapidly oligomerized and that NINJ1 oligomerization correlated with a loss of plasma membrane integrity. In line with a role for NINJ1 in executing ferroptosis-associated PMR, we found that deletion of *Ninj1* in macrophages or fibroblasts conferred prolonged protection against cell lysis but did not protect cells from death. Further analysis revealed that the absence of NINJ1 had no impact on the early events that are observable in ferroptotic cells, such as lipid peroxidation, calcium influx or cell swelling. However, NINJ1-deficiency delayed PI uptake and the release of small dextran from ferroptotic cells, both of which measure a loss of membrane integrity that is independent of cell lysis. Conversely, overexpression of NINJ1 accelerated the loss of membrane integrity. This indicates that NINJ1 not only causes PMR but is also essential for the initial loss of plasma membrane integrity during ferroptosis. Proteomic analysis of the supernatant from WT and *Ninj1*<sup>−/−</sup> cells revealed that NINJ1 was necessary for the release of cytosolic proteins and DAMPs from ferroptotic cells, suggesting that NINJ1-dependent lysis could be a driver of inflammation following ferroptosis induction. In summary, our data not only demonstrate that cell lysis during ferroptosis requires protein lesions formed by NINJ1, but also suggest a sequence of events for ferroptotic cell lysis, which starts with lipid peroxidation followed by opening of ion channels that then initiate an active destruction of the plasma membrane by NINJ1.

# Results

## NINJ1 oligomerizes during ferroptosis

NINJ1 has been reported to execute the cataclysmic PMR that is characteristic of many forms of lytic cell death, among them pyroptosis, late-stage apoptosis (secondary necrosis) and toxin-induced cell death (Kayagaki et al, 2021). A hallmark of NINJ1 activation is its oligomerization into plasma membrane assemblies of different size and shape (Degen et al, 2023). To begin exploring if NINJ1 executes PMR during ferroptosis, we first tested if CuOOH, an inducer of oxidative stress, or RSL3 and ML162, which inhibit GPX4-dependent reduction of peroxidized membrane lipids, induce ferroptosis in mouse bone marrow-derived macrophages (BMDMs). Treated BMDMs released LDH, a common indicator of PMR and cell lysis (Fig. 1A–C) and showed increased plasma membrane lipid peroxidation (Fig. 1D–F), measured using BODIPY™ 581/591 (C11-BODIPY), a dye whose fluorescence emission peak shifts upon oxidation of its polyunsaturated butadienyl portion (Pap et al, 1999). Ferrostatin-1 (Fer-1), a specific inhibitor of ferroptosis (Dixon et al, 2012) that scavenges alkoxyl radicals, efficiently blocked both lysis and lipid peroxidation, confirming that CuOOH, RSL3, and ML162 induce ferroptosis in BMDMs (Fig. 1A–F). LDH release was also efficiently blocked by Liproxstatin-1, a different ferroptosis inhibitor that targets lipid hydroperoxides (Fig. EV1A).

We next assessed the oligomerization of endogenous NINJ1 by immunofluorescence, upon treatment with these ferroptosis inducers (Fig. 1G). We observed that NINJ1 changed from a homogeneous plasma membrane staining to discrete plasma membrane clusters (or assemblies) (Fig. 1G, insets, arrows), which represent NINJ1 oligomers (Kayagaki et al, 2021; Degen et al, 2023). Similar NINJ1 oligomers formed upon treatment of BMDMs with the NLRP3 inflammasome activator nigericin, which induces pyroptosis (Fig. 1G, insets, arrows). Fer-1 blocked the formation of NINJ1 assemblies in cells treated with ferroptosis inducers but had no effect on nigericin-induced NINJ1 assemblies, confirming that NINJ1 oligomerization required initiation of ferroptosis. To further confirm that NINJ1 clusters represent oligomers, we performed crosslinking assays in WT BMDMs treated with CuOOH, RSL3, or ML162 (Fig. 1H–J), as done previously in pyroptotic cells (Degen et al, 2023). We found that upon ferroptosis induction, NINJ1 formed dimers, trimers, tetramers, and larger n-mers; and that the formation of these higher-order oligomers was strongly reduced by the treatment with Fer-1. On the contrary, we only detected monomers and some dimers in untreated cells. Importantly, Fer-1 did not directly block NINJ1 oligomerization, as it had no effect on nigericin-induced NINJ1 oligomerization (Fig. EV1B).

Next, we tested if NINJ1 also oligomerizes in human cells, by expressing human NINJ1 (hNINJ1) C-terminally tagged with green fluorescence protein (GFP) in HeLa cells and treating them with CuOOH. In untreated cells, NINJ1 was homogeneously distributed at the plasma membrane, whereas it clustered upon ferroptosis induction (Fig. 1K). To exclude the possibility that all plasma membrane proteins cluster upon lipid peroxidation and ferroptosis induction, we also analyzed HeLa cells expressing the transmembrane domain of Hemagglutinin tagged with GFP (HA<sup>TMD</sup>-GFP) (Fig. 1L). HA<sup>TMD</sup>-GFP remained plasma membrane localized upon CuOOH treatment and did not form any clusters. Moreover, we

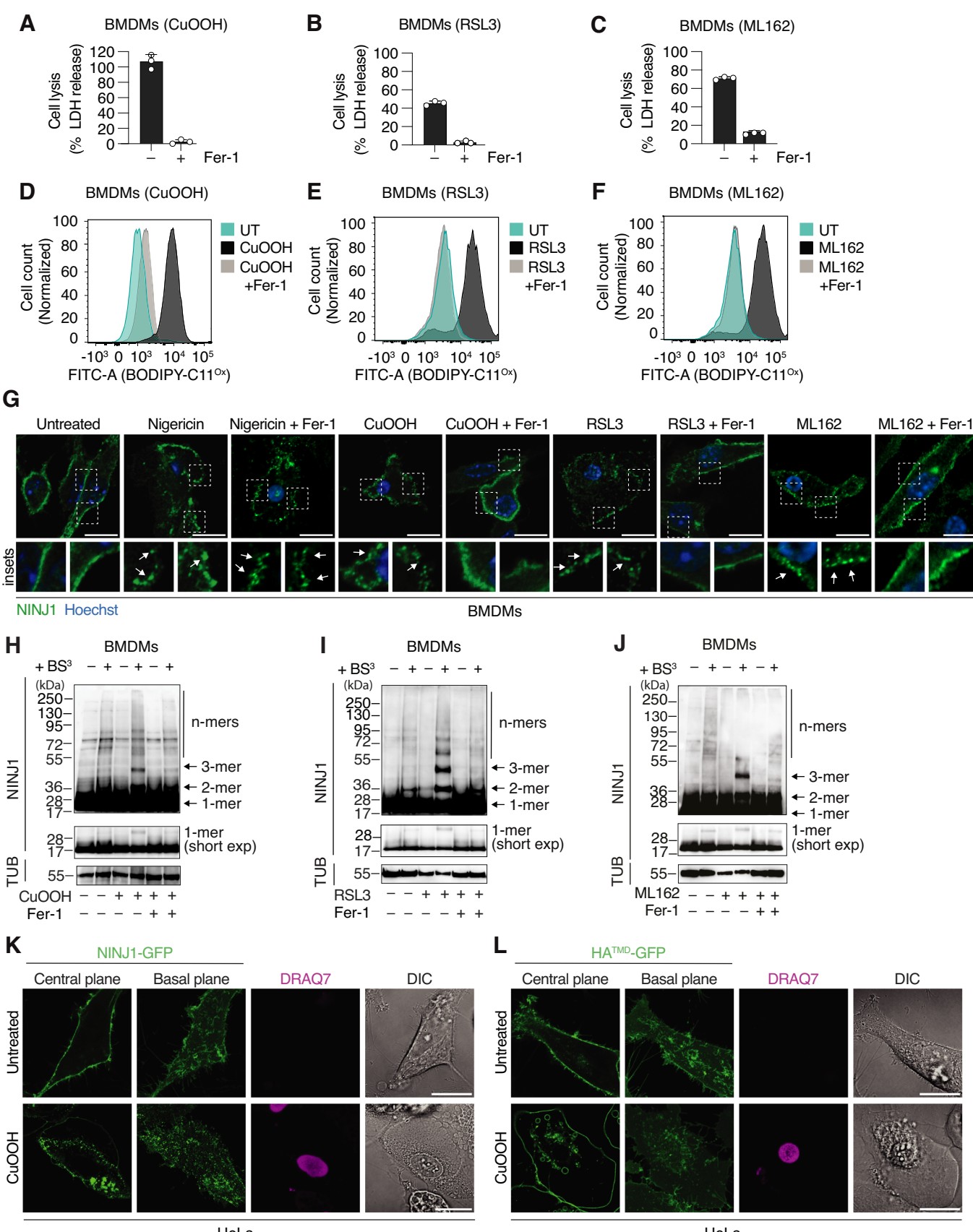

◄ **Figure 1.  Ferroptosis activators induce NINJ1 oligomerization.**

(A–C) Lactate dehydrogenase (LDH) release in WT BMDMs treated with 1 mM CuOOH for 5 h (A), 5 µM RSL3 for 6 h (B), or 5 µM ML162 for 5 h (C). (D–F) Histograms of BODIPY 581/591 C11 fluorescence at 540 nm emission wavelength, oxidized form (BODIPY-C11$^{Ox}$), showing cell counts in WT BMDMs left untreated (UT) or after 30 min of treatment with 1 mM CuOOH (D), 5 µM RSL3 (E), or 5 µM ML162 (F). (G) Immunofluorescence confocal microscopy of endogenous NINJ1 (green) in PFA-fixed WT BMDMs after treatment with 5µg/mL nigericin for 1.5 h, 1 mM CuOOH for 3 h, 5 µM RSL3 for 5 h or 5 µM ML162 for 5 h. White squares indicate inset images, and the arrows highlight NINJ1 oligomers. Scale bar: 10 µm. (H–J) Western blot analysis of endogenous NINJ1 in BMDMs treated with 1 mM CuOOH for 3 h (H) 5 µM RSL3 for 5 h (I) or 5 µM ML162 for 4 h (J) in the presence or absence of 25 µM Fer-1 followed by treatment with the membrane-impermeable crosslinker BS$^3$(+). Mixed supernatant and cell extracts were analyzed. Tubulin (TUB) serves as loading control. FL, full length; Short exp = short exposure; 1-mer = monomer; 2-mer = dimer; 3-mer = trimer and n-mer: higher-order oligomers. (K, L) Fluorescence confocal microscopy of HeLa cells expressing hNINJ1-GFP (K) or HA (Hemagglutinin)$^{TMD}$-GFP (L) treated with CuOOH for 3 h. Images show green fluorescence at the basal or central plane of the cell. DRAQ7 (maximum projection from a Z-stack), a membrane-impermeable DNA-binding dye, was used to track plasma membrane permeabilization. Scale bar: 20 µm. When indicated, 25 µM Fer-1 was added simultaneously with ferroptosis or pyroptosis activators (A–J). Data information: Graphs show the mean ± SD. Histograms show cell counts normalized to mode. Data are representative of three (A–F), two (H–J) or at least five (G, K, L) independent experiments performed in triplicate. Source data are available online for this figure.

confirmed that HeLa cells underwent cell lysis after treatment with CuOOH, RSL3 or ML162 (Fig. EV1C), although to a lesser degree than macrophages. In summary, these data show that ferroptosis induction in mouse and human cells triggers NINJ1 oligomerization into plasma membrane polymers that resemble the NINJ1 oligomers previously observed in pyroptotic or apoptotic cells (Degen et al, 2023).

## NINJ1 deficiency abrogates cell lysis and LDH release in ferroptotic BMDMs

To determine if NINJ1 contributes to PMR and cell lysis in ferroptotic cells, we measured LDH release from WT and *Ninj1$^{-/-}$* BMDMs treated with ferroptosis inducers. *Ninj1*-deficiency significantly reduced LDH release upon treatment with CuOOH, RSL3 or ML162 (Fig. 2A–C) but had no effect on the basal levels of LDH release or detergent induced lysis (Fig. EV2A), indicating that NINJ1 causes PMR upon ferroptosis induction in BMDMs. To further confirm that NINJ1 controlled PMR during ferroptosis, we analyzed the morphology of WT and *Ninj1$^{-/-}$* BMDMs treated with ferroptosis inducers (Fig. 2D). We found that both genotypes presented membrane blebs and formed larger membrane balloons, which is a characteristic feature of cells undergoing necrotic cell death (Kayagaki et al, 2021), such as ferroptosis or pyroptosis (Fig. 2D, arrows). However, ferroptotic WT cells lost the ballooning morphology after 5 h, which indicates that they underwent PMR, while *Ninj1$^{-/-}$* BMDMs retained the ballooning morphology at the same time point.

Recently, we showed that during pyroptosis NINJ1 oligomerizes into branched and filamentous assemblies in the plasma membrane of dying cells, and solved the cryo-EM structure of filamentous NINJ1 (Degen et al, 2023). To confirm that NINJ1 filament formation was also required for ferroptosis-associated PMR, we retrovirally transduced *Ninj1$^{-/-}$* BMDMs with vectors expressing NINJ1$^{WT}$, NINJ1$^{K45Q}$, or NINJ1$^{D53A}$. Of note, residues K45 and D53 form an important salt bridge that stabilizes the interaction between individual protomers in the NINJ1 filament, and mutating these residues completely abrogates filament formation *in vitro* (Degen et al, 2023). We first confirmed that WT and mutant NINJ1 were expressed to similar levels (Fig. EV2B) and that the proteins localized to the plasma membrane (Fig. EV2C,D), before analyzing cell lysis following ferroptosis induction with CuOOH (Fig. 2E). We found that BMDMs expressing NINJ1$^{WT}$ released significantly higher levels of LDH after 2 h of CuOOH treatment than the vector control, but that in contrast, LDH release from cells expressing

NINJ1$^{K45Q}$ and NINJ1$^{D53A}$ was comparable to the vector control. LDH levels at this time point were comparable to non-transduced controls (Fig. 2F). This confirmed that ferroptosis-associated PMR requires NINJ1 filament formation.

Next, we asked if NINJ1 deficiency protects ferroptotic cells from cell death. By measuring ATP levels in CuOOH-treated WT and *Ninj1$^{-/-}$* BMDMs, we found that viability in both genotypes declined with comparable kinetics (Fig. 2G). Thus, the absence of NINJ1 did not protect the cells against cell death, which implies that cell lysis is not the cause but rather one consequence of ferroptotic cell death. Finally, we determined for how long *Ninj1*-deficiency protects ferroptotic cells from undergoing lysis. We performed a time course of LDH release up to 14 h post CuOOH treatment and showed that *Ninj1$^{-/-}$* BMDMs failed to lyse even at late time points (Figs. 2H and EV2E), similar to what has been observed in pyroptotic cells (Kayagaki et al, 2021). In summary, our results show that NINJ1 not only oligomerizes during ferroptosis, but it is the cause of ferroptosis-associated PMR in macrophages.

## NINJ1-driven cell lysis during ferroptosis is independent of pyroptotic or necroptotic cell death executioners

Unlike many cell lines that are commonly used to study ferroptosis, BMDMs can activate the NLRP3 inflammasome pathway in response to a disruption in plasma membrane integrity. To exclude that the observed NINJ1-dependent PMR is a consequence of NLRP3 inflammasome activation upon ferroptosis induction, we compared LDH release in WT and *Nlrp3$^{-/-}$* BMDMs after treatment with CuOOH, RSL3, or ML162 (Fig. EV3A–C). *Nlrp3$^{-/-}$* BMDMs displayed comparable LDH release levels as WT BMDMs after ferroptosis induction, thus demonstrating that the NLRP3 inflammasome was not required for ferroptotic cell lysis. In addition, we also examined BMDMs lacking GSDMD, which gets activated downstream of inflammatory caspases and drives NINJ1 activation, and GSDME that induces pyroptosis downstream of apoptotic caspases (Rogers et al, 2017). *Gsdmd*-deficiency did not impact LDH release levels after treatment with the ferroptosis inducers (Fig. EV3D–F). On the other hand, *Gsdme*-deficiency induced a slight reduction of LDH release upon treatment with ML162 (Fig. EV3F), yet not comparable to *Ninj1*-deficiency (Fig. 2C). Finally, we tested if necroptosis induction could account for NINJ1 activation in ferroptotic BMDMs, as NINJ1 contributes partially to PMR in cells undergoing this form of cell death. However, BMDMs incapable of initiating (*Ripk3$^{-/-}$/Casp8$^{-/-}$*) or executing necroptosis (*Mlkl$^{-/-}$*) released similar amounts of LDH as

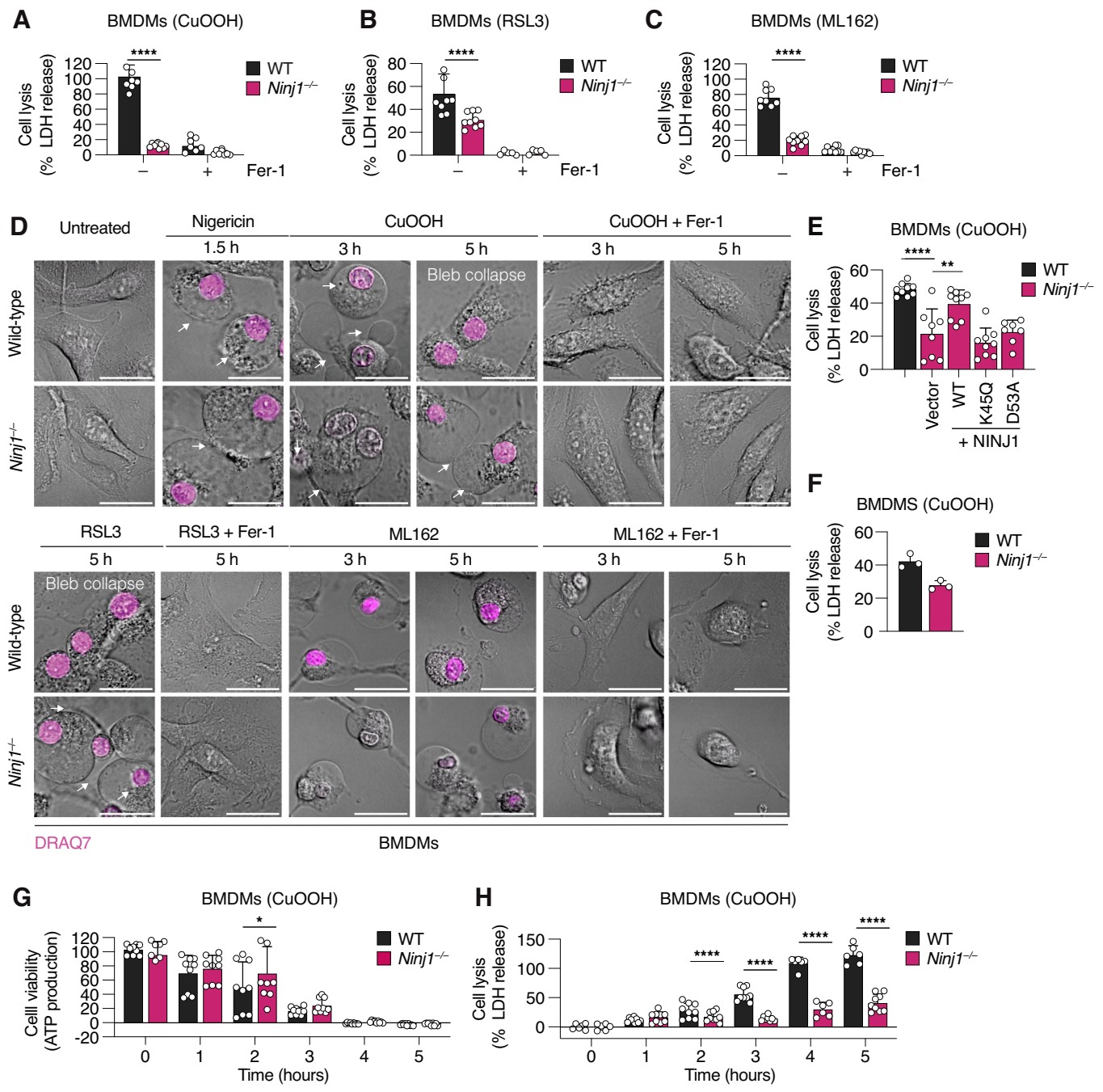

**Figure 2. *Ninj1*-deficiency blocks cell lysis and LDH release in ferroptotic BMDMs.**

(A–C) LDH release in WT and *Ninj1*⁻/⁻ BMDMs treated with 1 mM CuOOH for 5 h (A), 5 µM RSL3 for 6 h (B), or 5 µM ML162 for 5 h (C). (D) Confocal microscopy of WT and *Ninj1*⁻/⁻ BMDMs after treatment with nigericin 5µg/mL for 1.5 h, 1 mM CuOOH for 3 h or 5 h, 5 µM RSL3 for 5 h or 5 µM ML162 for 3 h or 5 h. Cells were imaged without fixation to show BMDMs morphology, plasma membrane blebs and balloons (white arrows) and bleb collapse. DRAQ7, a membrane-impermeable DNA-binding dye, was used to track plasma membrane permeabilization. Scale bar: 20 µm. (E) LDH release in WT or *Ninj1*⁻/⁻ BMDMs reconstituted with a retroviral vector expressing WT or mutant NINJ1 and treated with 1 mM CuOOH for 2 h. Reconstitution with a GFP-expressing vector was used as a control. (F) LDH release in WT or *Ninj1*⁻/⁻ BMDMs treated with 1 mM CuOOH for 2 h. (G, H) Cell viability (G), measured based on the absorbance of intracellular ATP signal, and LDH release (H), in WT and *Ninj1*⁻/⁻ BMDMs upon treatment with 1 mM CuOOH for 1, 2, 3, 4, or 5 h. When indicated, 25 µM Fer-1 was added simultaneously with ferroptosis and pyroptosis activators (A–D). Data information: All graphs show the mean ± SD. Data are representative of two (D) or three (F) independent experiments (D) or pooled from three independent experiments performed in triplicate (A–C, E, G, H). Statistical analysis was done using two-way ANOVA (A–C) or one-way ANOVA with multiple comparisons tests (E, G, H). *$P < 0.05$; **$P < 0.01$; ***$P < 0.001$; ****$P < 0.0001$. Source data are available online for this figure.

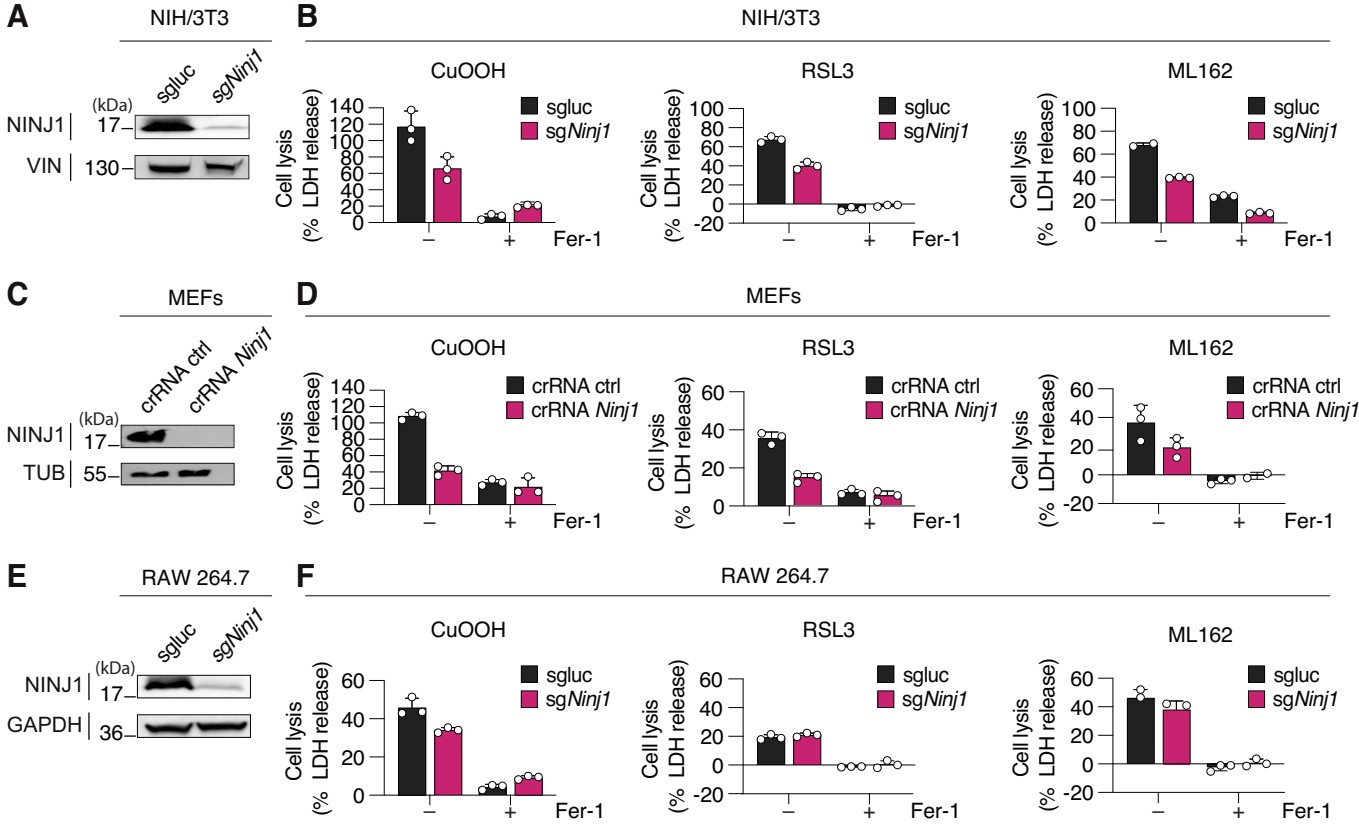

**Figure 3. NINJ1 controls LDH release from ferroptotic fibroblast and partially from RAW 264.7 cells.**

(A) Western blot analysis of NINJ1 expression in (sgluc) and *Ninj1* knockout (sg*Ninj1*) NIH/3T3. (B) LDH release in WT (sgluc) and *Ninj1* knockout (sg*Ninj1*) NIH/3T3 after treatment with 1 mM CuOOH for 5 h, 5 µM RSL3 for 8 h, or 5 µM ML162 for 8 h. (C) Western blot analysis of NINJ1 expression in WT (crRNA ctrl) and *Ninj1* knockout (crRNA *Ninj1*) MEFs. (D) LDH release in WT (crRNA ctrl) and *Ninj1* knockout (crRNA *Ninj1*) MEFs after treatment with 0.4 mM CuOOH for 5 h, 5 µM RSL3 for 8 h, or ML162 5 µM for 8 h. (E) Western blot analysis of NINJ1 expression in WT (sgluc) and *Ninj1* knockout (sg*Ninj1*) RAW 264.7. (F) LDH release in WT (sgluc) and *Ninj1* knockout (sg*Ninj1*) RAW 264.7 after treatment with 1 mM CuOOH for 2 h, 5 µM RSL3 for 3 h, or 5 µM ML162 for 2 h. Cells were lentivirally transduced with Cas9 and a sgRNA targeting luciferase (sgluc) or crRNA (crRNA ctrl), used as controls, or NINJ1 (sg*Ninj1* and crRNA *Ninj1*). Cell extracts were analyzed (A, C, E). Vinculin (VIN), Tubulin (TUB), or GAPDH are loading controls. When indicated, 25 µM Fer-1 was added simultaneously with ferroptosis activators (B, D, F). Data information: All graphs show the mean ± SD. Data are representative of three independent experiments performed in triplicate. Source data are available online for this figure.

WT BMDMs (Fig. EV3G–I). In summary, these data exclude that NINJ1 activation in CuOOH-, RSL3-, or ML162- treated BMDMs is caused by an activation of an alternative cell death program downstream of ferroptosis triggers.

## NINJ1 controls ferroptosis-induced cell lysis in fibroblasts

Most reports have so far shown a role for NINJ1 in macrophages, but its role in other cell types remains largely untested. We therefore generated different *Ninj1*-deficient cell lines, using CRISPR/Cas9 gene targeting and performed LDH analysis following a kinetic analysis at different concentrations of ferroptosis activators (Appendix Fig. S1). We observed that upon treatment with CuOOH, RSL3 and ML162 *Ninj1*-deficient NIH/3T3 cells (murine embryonic fibroblasts) showed a robust reduction in LDH release (Fig. 3A,B; Appendix Fig. S1A). Similarly, we found that *Ninj1*-deficiency reduced cell lysis in MEFs (murine embryonic fibroblasts) upon treatment with different concentrations of the three ferroptosis inducers (Fig. 3C,D; Appendix Fig. S1B).

Interestingly, we found that absence of NINJ1 caused only a mild reduction in cell lysis in RAW264.7 macrophages (Fig. 3E,F; Appendix Fig. S1C), in line with a previous report (Hirata et al, 2023). By contrast, BMDMs showed almost complete dependance on NINJ1 for cell lysis over a range of concentrations of ferroptosis activators (Appendix Fig. S1D). While these results suggest that NINJ1 is essential in several different cell types, they nevertheless reveal that the importance of NINJ1 for cell lysis varies considerably due to factors yet unknown. Of note, each activator induced cell lysis with different kinetics depending on the cell type.

## During ferroptosis, NINJ1 is activated downstream of lipid peroxidation and Ca²⁺ influx

A hallmark of ferroptosis is the generation of lipid peroxides in the plasma membrane, which is then followed by an increase in cytosolic $Ca^{2+}$ levels, cell rounding and loss of membrane integrity (Hirata et al, 2023). To determine which of these steps depend on NINJ1, we stained WT or $Ninj1^{-/-}$ BMDMs with BODIPY™ 581/591 C11 and either left them untreated or treated them with

CuOOH +/- Fer-1 before quantifying the percentage of oxidized BODIPY (BODIPY$^{Ox}$)-positive cells at different time points post-treatment. CuOOH induced an increase in BODIPY$^{Ox}$-positive cells within 30 min of treatment in both WT or $Ninj1^{-/-}$ BMDMs, which was maintained at later time points (Fig. 4A,B), indicating that in ferroptotic cells NINJ1 acts downstream of lipid peroxidation. Similar results were obtained for cells treated with RSL3 or ML162 (Fig. 4C,D).

Recent reports have shown that lipid peroxidation is followed by an increased permeability of the plasma membrane to mono- and divalent cations Na$^+$, K$^+$, and Ca$^{2+}$, and that this precedes cell lysis (Pedrera et al, 2021; Riegman et al, 2020; Hirata et al, 2023). These ion fluxes are partially mediated by the opening of mechanosensitive plasma membrane channels, such as Piezo-1 and TRP channel (Hirata et al, 2023). We thus investigated if NINJ1 contributed to ferroptosis-associated ion fluxes, or if NINJ1 was activated downstream of ion fluxes by measuring Ca$^{2+}$ influx in WT and $Ninj1^{-/-}$ BMDMs using the fluorescent calcium probe Fluo-8 AM, an intracellular dye that increases in fluorescence upon binding to calcium. CuOOH treatment increased the Fluo-8 signal in both genotypes with comparable kinetics (Fig. 4E), indicating that NINJ1 had no impact on channel opening. However, we observed that the Fluo-8 signal peaked and rapidly declined in WT cells at around 90 min post-treatment, while it peaked at higher level and later time points in $Ninj1^{-/-}$ BMDMs, most likely as $Ninj1$-deficient cells maintain membrane integrity for longer. In summary, these experiments suggest that early ferroptotic events, such as lipid peroxidation and the influx of calcium that is caused by membrane channel opening, are independent of NINJ1.

## Ferroptosis-induced membrane permeabilization requires NINJ1

Ferroptosis involves the formation of protein- or lipid-based nanopores downstream of lipid peroxidation and Ca$^{2+}$ influx, which initially cause a loss of plasma membrane integrity (PMI) (measured by uptake of DNA-binding dyes) and at a later stage cell lysis (i.e., LDH release) (Pedrera et al, 2021). We asked if NINJ1 forms these nanopores, and if it controls both events or if another protein causes loss of PMI upstream of NINJ1. This would be analogous to GSDMD pores, which cannot cause PMR themselves but promote the uptake of small membrane impermeable DNA-binding dyes, such as PI (668 Da) or DRAQ7 (400 Da) (Kayagaki et al, 2021; Degen et al, 2023).

We thus compared the role of NINJ1 in PI uptake in pyroptotic and ferroptotic BMDMs. PI uptake after treatment with the inflammasome activator nigericin was comparable between WT and $Ninj1^{-/-}$ cells (Fig. 5A), in line with previous reports that showed that GSDMD pores are sufficient for the induction of a loss of PMI. By contrast, we found that $Ninj1$-deficiency delayed PI uptake in BMDMs treated with different concentrations of CuOOH, RSL3, or ML162 (Figs. 5B–D and EV4A). Consistently, microscopy analysis showed reduced levels of PI-positive cells in CuOOH-treated $Ninj1^{-/-}$ BMDMs (Fig. EV4B). $Ninj1$-deficiency also delayed PI uptake in MEFs treated with CuOOH, RSL3, or ML162, indicating that this effect was not restricted to macrophages (Fig. EV4C). Similarly, PI uptake was delayed in $Ninj1$-deficient NIH/3T3 and RAW 264.7 cells (Appendix Fig. S2). Fer-1 treatment abrogated PI uptake in all cases, confirming that loss of

PMI is driven by ferroptosis induction. Previous work had shown that GSDMD pores can also mediate the release of small fluorescent dextran from pyroptotic BMDMs, even in absence of NINJ1 (Kayagaki et al, 2021). We thus preloaded WT or $Ninj1^{-/-}$ BMDMs with fluorescent dextran (3 kDa dextran, Alexa Fluor™ 488) and measured the decline in dextran-positive cells after ferroptosis induction. This analysis showed that 3 kDa dextran was released faster from WT BMDMs than from $Ninj1^{-/-}$ BMDMs upon ferroptosis induction (Fig. 5E–G), further supporting that NINJ1 is the driver of plasma membrane integrity loss in ferroptotic cells.

We next asked how NINJ1 oligomerization correlated with the loss of membrane integrity by treating HeLa cells expressing NINJ1-GFP with CuOOH in the presence of DRAQ7, a marker for membrane integrity. In live cells, NINJ1 was homogeneously distributed at the plasma membrane (Fig. 5H; Appendix Fig. S3A, C and Movie EV1), and rapidly clustered upon treatment with CuOOH, as determined by quantitative inhomogeneity analysis of the plasma membrane hNINJ1-GFP signal (Fig. 5I) (Paparelli et al, 2016). Interestingly, NINJ1 oligomerization preceded DRAQ7 influx by around 10–20 min in the inhomogeneity analysis, which is strikingly different from what we had previously observed in pyroptotic cells, where NINJ1 oligomerization and DRAQ7 influx happen simultaneously (Degen et al, 2023). This again supports the notion that in ferroptotic cells NINJ1 itself causes membrane integrity loss, while in pyroptotic cells the loss of membrane integrity is a consequence of GSDMD pore formation.

To confirm that clustering is not a general feature of plasma membrane proteins in ferroptotic cells, we also analyzed HeLa cells expressing HA$^{TMD}$-GFP (Fig. 5J; Appendix Fig. S3B, D and Movie EV2). While we did not observe any clustering of HA$^{TMD}$ upon ferroptosis induction, we found that DRAQ7 uptake proceeded with slower kinetics than in HeLa cells expressing NINJ1-GFP. This difference in DRAQ7 is most likely caused by the fact that overexpression of NINJ1 accelerates the loss of membrane integrity in these cells. In summary, these data support the conclusion that during ferroptosis NINJ1 activation is not the consequence of plasma membrane permeabilization caused by a gasdermin-like pore, but that NINJ1 itself causes both loss of plasma membrane integrity and subsequent cell rupture during ferroptosis.

## NINJ1 mediates the release of DAMPs from ferroptotic cells

Since ferroptosis is characterized by the release of intracellular contents upon PMR (Tang et al, 2021), we aimed to determine the fraction and identity of the host proteome that is released in a NINJ1-dependent manner (i.e., the ferroptotic secretome). To do so, we employed Stable Isotope Labeling using Amino acids in Cell culture (SILAC) and compared proteins in the supernatants of WT and $Ninj1^{-/-}$ iBMDMs after CuOOH treatment. We ensured that cell lysis was dependent on NINJ1 in iBMDMs (Appendix Fig. S4A). In both WT and $Ninj1^{-/-}$ cells, the distribution of protein enrichments was shifted to values larger than zero (Fig. 6A-I,A-II) when comparing treated to untreated cells, indicating an increased release upon induction of ferroptosis in both genotypes. However, the overall number of proteins released from $Ninj1^{-/-}$ iBMDMs was lower than the number released from WT cells, and when directly comparing WT treated with $Ninj1^{-/-}$ treated cells we

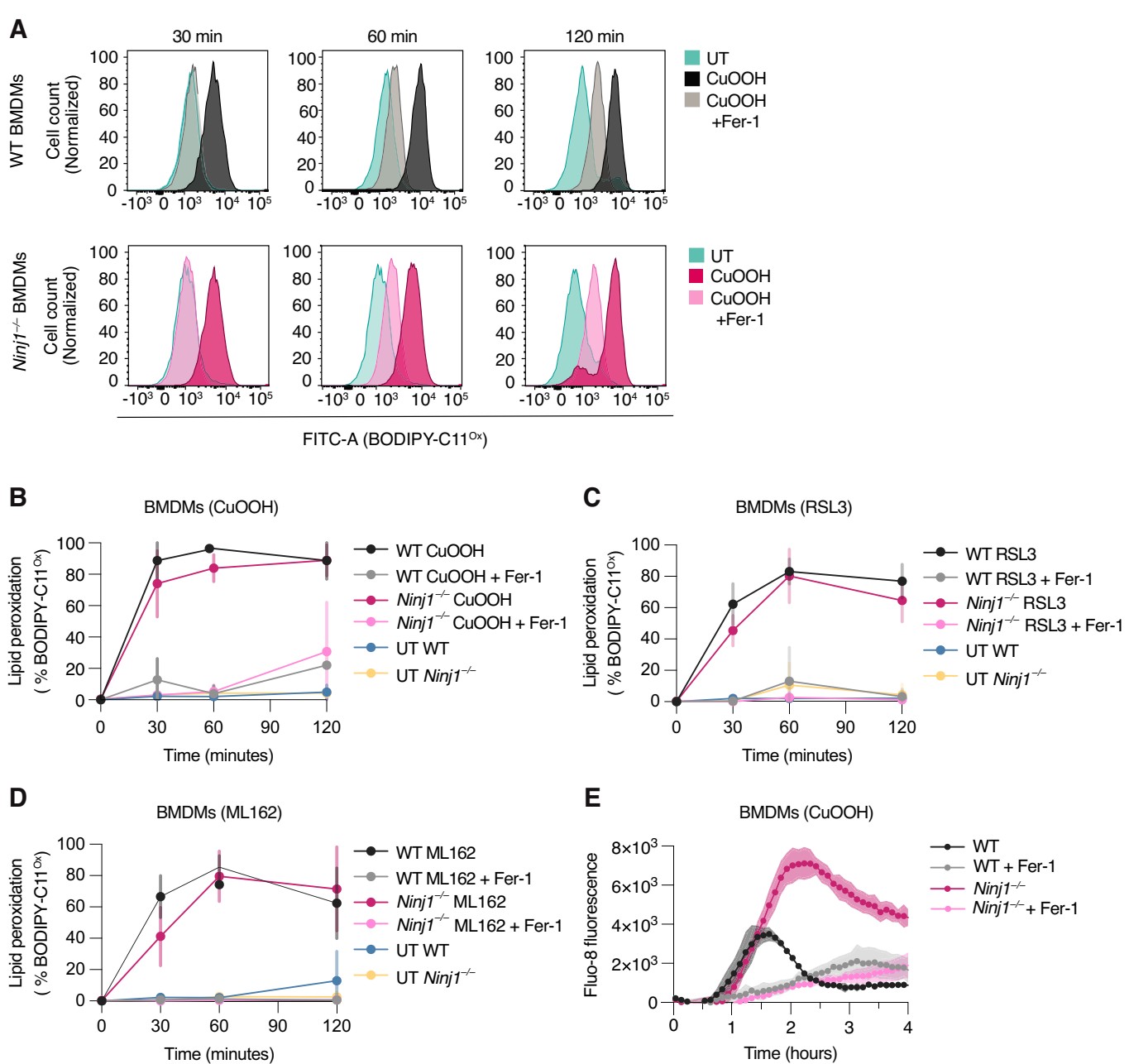

**Figure 4. NINJ1 is activated downstream of lipid peroxidation and Ca²⁺ influx in BMDMs.**

(A) Representative analysis of BODIPY 581/591 C11 fluorescence at 540 nm emission wavelength (oxidized form, BODIPY-C11^Ox). Histograms show normalized cell counts in WT and *Ninj1*⁻/⁻ BMDMs left untreated or after treatment with 1 mM CuOOH for 30, 60, or 120 min. (B) Quantification of (A) with additional replicates included. Percentage of WT and *Ninj1*⁻/⁻ BMDMs positive for BODIPY-C11^Ox in untreated cells (UT) or after treatment with 1 mM CuOOH for 30, 60, or 120 min. (C) Percentage of WT and *Ninj1*⁻/⁻ BMDMs positive for BODIPY-C11^Ox in untreated cells (UT) or after treatment with 5 μM RSL3 for 30, 60, or 120 min. (D) Percentage of WT and *Ninj1*⁻/⁻ BMDMs positive for BODIPY-C11^Ox in untreated cells (UT) or after treatment with 5 μM ML162 for 30, 60, or 120 min. (E) Time course analysis from 0–4 h of Fluo-8 (intracellular Ca²⁺ probe) fluorescence signal in WT and *Ninj1*⁻/⁻ BMDMs treated with 1 mM CuOOH. When indicated, 25 μM Fer-1 was added simultaneously with ferroptosis activators (A–E). Data information: Graphs show the mean ± SD. Histograms show cell counts normalized to mode. Data are representative of three independent experiments performed in triplicate (A, E) or pooled from three independent experiments performed in triplicate (B–D). Source data are available online for this figure.

observed an overall shift in enrichment values towards the WT treated condition (Fig. 6AIII, median logarithmic enrichment: 1.347). We next compared the enrichments for treated WT iBMDMs with respect to either untreated WT or treated *Ninj1*⁻/⁻, and found a strong correlation between the two conditions

(Pearson R: 0.70, Appendix Fig. S4B). This suggests that the profile of proteins that are released from CuOOH-treated *Ninj1*⁻/⁻ cells strongly resembles the WT untreated condition, implying that NINJ1 controls the release of the majority of proteins found in the secretome of ferroptotic cells.

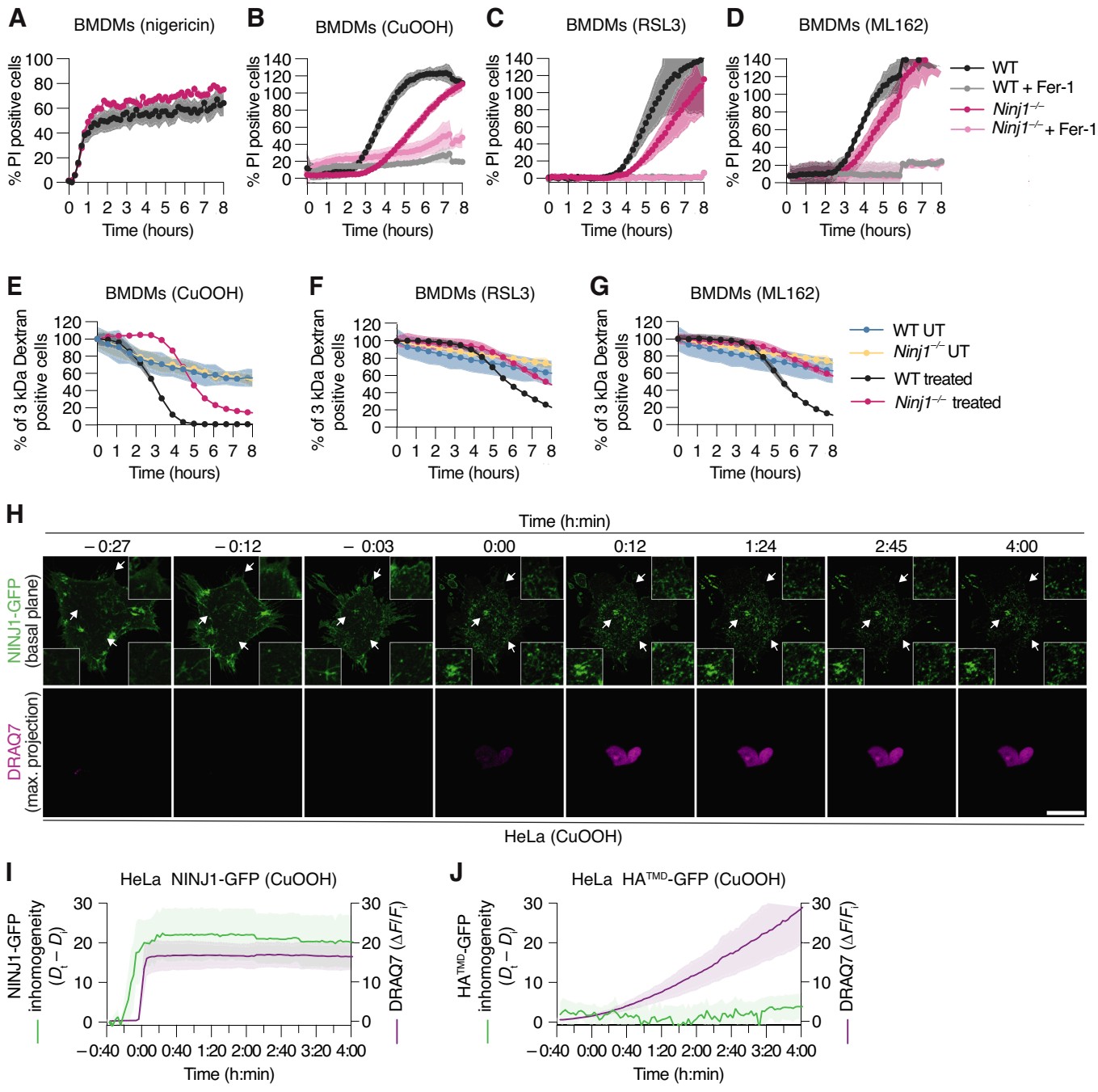

**Figure 5.  NINJ1 controls plasma membrane permeabilization during ferroptosis in macrophages and fibroblasts.**

(A–D) Percentage of propidium iodide (PI, Mw = 668 Da) uptake in WT and *Ninj1*$^{-/-}$ BMDMs over time (0–8 h) after treatment with 5 μg/mL nigericin (A), 1 mM CuOOH (B), 5 μM RSL3 (C), or 5 μM ML162 (D). When indicated, 25 μM Fer-1 was added simultaneously with ferroptosis activators. (E–G) Release of 3 kDa fluorescent dextran from WT and *Ninj1*$^{-/-}$ BMDMs left untreated (UT) or treated with 1 mM CuOOH (E), 5 μM RSL3 (F), or 5 μM ML162 (G) over time (0–8 h). (H) Time-lapse fluorescence confocal microscopy of HeLa cells expressing hNINJ1-GFP after 1 mM CuOOH treatment. Images show the green fluorescence at the basal plane of the cell and the influx of DRAQ7 (maximum projection from a Z-stack) to track plasma membrane permeabilization. White arrows indicate regions that are enlarged in the insets. Time was normalized to the onset of increase in DRAQ7 nuclear fluorescence. Scale bar: 20 μm. (I, J) Normalized quantification of the inhomogeneity distribution of NINJ1-GFP (I) or HA$^{TMD}$-GFP (J) at the basal plane of cells, and DRAQ7 nuclear fluorescence intensity after 1 mM CuOOH treatment over time. The inhomogeneity distribution at each time point ($D_t$) was normalized to the distribution inhomogeneity at the initial time point of the experiment ($D_i$). Data information: All graphs show the mean ± SD. Data are representative of two (A, E) or three (F, G) independent experiments performed in triplicate, representative of 8 independent experiments (H), or pooled from two independent experiments performed in triplicate (B–D) or 8 (I) or 6 (J) independent experiments. Source data are available online for this figure.

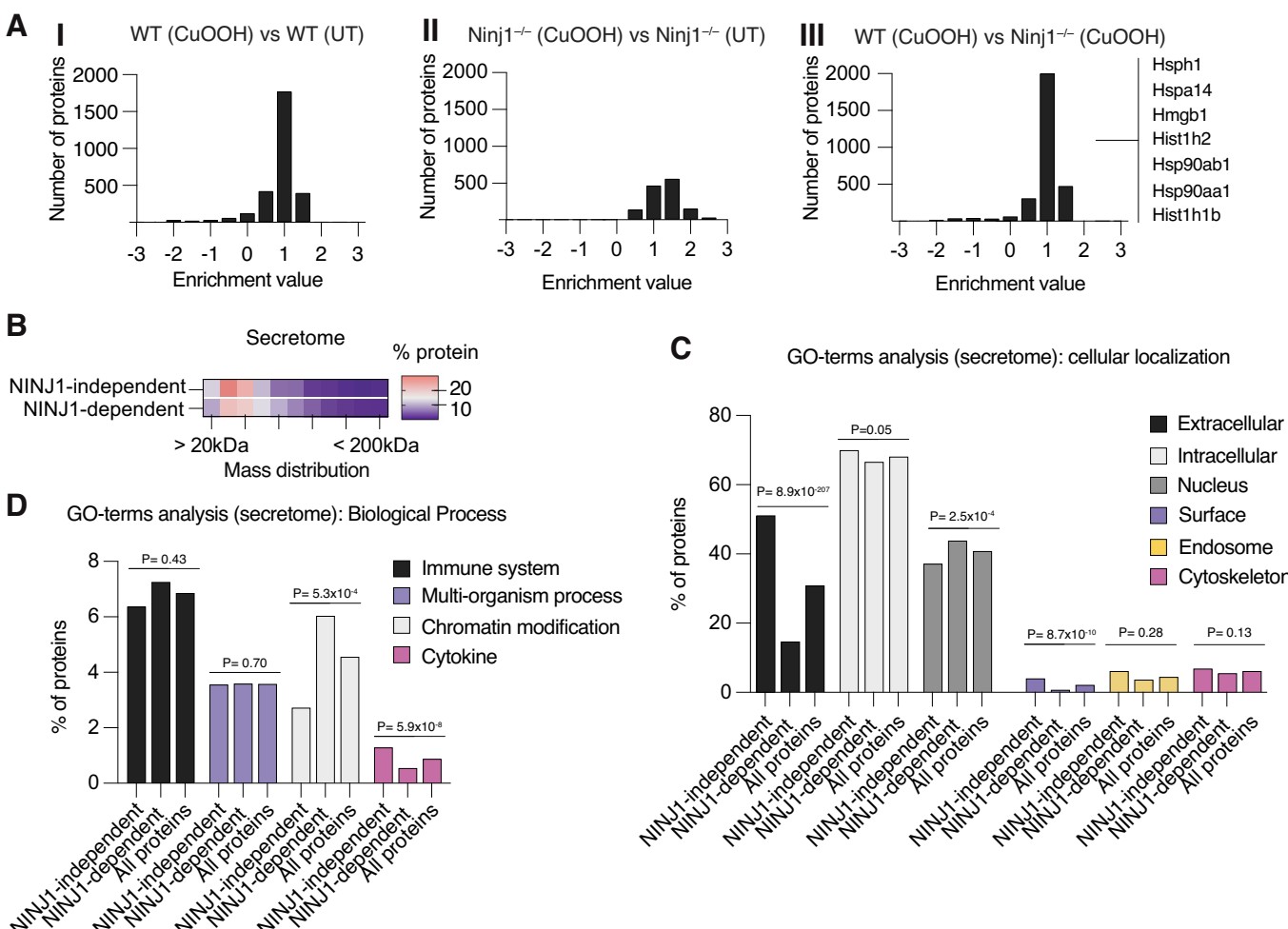

**Figure 6.  NINJ1 controls the release of DAMPs from ferroptotic cells.**

(A) SILAC enrichment distributions, comparing: WT iBMDMs treated with 1 mM CuOOH to untreated (UT) (A I), *Ninj1⁻/⁻* iBMDMs treated with 1 mM CuOOH to untreated (UT) (A II), and WT iBMDMs treated with 1 mM CuOOH to *Ninj1⁻/⁻* iBMDMs treated with 1 mM CuOOH (A III). CuOOH treatment for 2 h. Selected DAMPs released in a NINJ1-dependent manner upon ferroptosis induction are indicated (full list in Dataset EV1). (B) Heatmap depicting the distribution of protein sizes detected with a logarithmic enrichment larger than 0.5 in WT and *Ninj1⁻/⁻* iBMDMs treated with 1 mM CuOOH for 2 h. Bin centers were chosen in 20 kDa increments, ranging from 10 kDa (i.e., 0 to 20 kDa) to 210 kDa (i.e., larger than 200 kDa). (C, D) Cellular Localization (C) and Biological Process (D) GO-term analyses of the NINJ1-dependent (WT iBMDMs) and NINJ1-independent (*Ninj1⁻/⁻* iBMDMs) ferroptotic secretome from iBMDMs treated with 1 mM CuOOH for 2 h. Bars depict the percentage of proteins that are annotated with a given GO-term among all proteins identified in the sample. Data information: *p*-values were calculated by Chi-square test, assuming an even distribution. Data are pooled from two independent experiments (A–D), requiring presence of a given protein in both datasets to ensure reliability.

Since some proteins were still released in ferroptotic *Ninj1⁻/⁻* iBMDMs, we wondered if their characteristics could explain their release in a NINJ1-independent manner. A mass distribution analysis showed that the proteins released in a NINJ1-independent manner were smaller in comparison to the average size of the NINJ1-dependent secretome (Fig. 6B), indicating that the absence of NINJ1 imposes an overall size restriction for release. Moreover, by evaluating GO-terms we found that the NINJ1-independent secretome was more likely to be associated with the extracellular compartment or the cell surface, as determined by Chi-square test (Fig. 6C), rather than intracellular compartments like cytoskeleton or nucleus. Similarly, we found that the NINJ1-independent secretome was associated with specific biological functions, such as cytokines (Fig. 6D), while the NINJ1-dependent secretome was associated with intracellular

functions such as chromatin modification. As this indicated that the proteins released in a NINJ1-dependent manner were mostly cytosolic or nuclear proteins, we investigated whether commonly known DAMPs are released by NINJ1-dependent PMR (Roh and Sohn, 2018). Interestingly, a large fraction of well-described DAMPs, including HMGB1, several histones, actin-related proteins, and heat-shock proteins were specifically enriched in treated WT cells (Fig. 6A; Dataset EV1), indicating that their release during ferroptosis requires NINJ1. Next, we transferred supernatants of CuOOH-treated WT and *Ninj1⁻/⁻* BMDMs onto WT BMDMs and found an upregulation of *Tnf* and *Il-1b* RNA levels that was dampened by the absence of NINJ1 (Appendix Fig. S4C,D). In summary, this analysis shows that DAMP release from ferroptotic macrophages largely proceeds via NINJ1-dependent PMR.

# Discussion

The exact mechanism by which lipid peroxidation results in the lysis of ferroptotic cells still remains incompletely understood. It was recently reported that calcium influx is one of the first events that follows lipid peroxidation (Pedrera et al, 2021; Riegman et al, 2020; Hirata et al, 2023), and that it precedes the uptake of DNA-binding dyes, which indicate a loss of membrane integrity, and LDH release, a marker of PMR and cell lysis. Calcium influx was also shown to be caused by the opening of mechanosensitive cation channels, such as Piezo-1 and TRP channels, that respond to increases in membrane tension caused by lipid peroxidation (Hirata et al, 2023).

Given that cell lysis was found to be blocked by osmoprotectants and that opening of these channels collapses monovalent cation gradients ($Na^+$, $K^+$) and drives cell volume increases, it was proposed that ferroptotic cell rupture is caused by increased osmotic pressure (Riegman et al, 2020; Pedrera et al, 2021; Hirata et al, 2023). However, here we demonstrate that while ferroptotic cell lysis occurs downstream of ion fluxes and cells swelling, it is nevertheless an active process that is carried out by the plasma membrane protein NINJ1.

Our results show that NINJ1 oligomerizes into plasma membrane assemblies during ferroptosis and that NINJ1 is required for late ferroptotic events, i.e., the uptake of small DNA-binding dyes and cell lysis. By contrast, NINJ1-deficiency has no impact on early ferroptotic events, such as lipid peroxidation, the opening of plasma membrane channels and cell swelling. These findings, together with studies published by others, allows us to propose the following model for ferroptotic cell lysis (Fig. EV5): (1) Ferroptosis-induced cell lysis starts with the formation of lipid peroxides at the plasma membrane which cause and increase in membrane tension. (2) The increased membrane tension causes the opening of membrane channels and non-selective ion fluxes leading to ion disbalance. This results in an increase in the cell volume, which is visible by the appearance of membrane blebbing and ballooning, and an overall swollen morphology. (3) Following this step NINJ1 is activated, possibly because of ion disbalance, swelling or a yet unrecognized signal, and oligomerizes to form amphipathic filaments that form lesions within the plasma membrane. These lesions remain initially limited in size and allow only the entry of small molecules. (4) Eventually NINJ1 lesions cause complete membrane rupture and cell lysis. As a consequence of NINJ1-dependent PMR, ferroptotic cells start to leak intracellular contents among them many DAMPs that can drive inflammation (Tang et al, 2021).

NINJ1 has previously been reported to execute PMR during other forms of programmed cell death, most notably during pyroptosis where it is activated downstream of the formation of GSDMD pores (Kayagaki et al, 2021). Comparing viability, membrane integrity and cell lysis of pyroptotic and ferroptotic cells, we find that NINJ1 plays differential roles in these forms of cell death. While NINJ1 is required for LDH release in both instances, it is dispensable for the loss of membrane integrity in pyroptotic cells (Kayagaki et al, 2021; Degen et al, 2023). The loss of membrane integrity of pyroptotic cell is completely dependent on GSDMD pores, which due to their large size (22 nm inner diameter (Xia et al, 2021)) allow the uptake of the DNA-binding dyes DRAQ7 or PI (400 Da and 668 Da, respectively), and the release of small dextrans (3 kDa). By contrast, both DNA dye uptake and

dextran release is strongly NINJ1-dependent during ferroptosis, indicating that during ferroptosis, there is not the formation of a GSDMD-like large pore and that the formation of NINJ1 lesions directly follows channel opening and cell swelling. Delayed PI uptake and dextran release eventually occurs in ferroptotic $Ninj1^{-/-}$ BMDMs, even though they do not undergo PMR even after several hours of incubation with CuOOH, suggesting the formation of some small lesions or pores that cannot induce membrane rupture. It remains to be investigated whether these are formed by other proteins being activated or represent a loss of membrane integrity caused by the accumulation of peroxided lipids (Bour et al, 2019; Agmon et al, 2018).

One limitation of our study is the observed inducer- and cell type-specific differences. In RAW 264.7 macrophages $Ninj1$-deficiency did not provide a significant protection against PMR upon ferroptosis induction, consistent with a previous study (Hirata et al, 2023). Also, we observed that RSL3-induced cell lysis was less NINJ1-dependent than cell lysis induced by CuOOH or ML162. Since different ferroptosis inducers act by distinct mechanisms and induce cell death with variable kinetics in different cell lines (Feng and Stockwell, 2018; Pedrera et al, 2021), we speculate that intrinsic differences in the extent and speed of lipid peroxidation and the resulting instability of the plasma membrane could account for whether NINJ1 contributes to PMR. Thus, the role of NINJ1 for cell lysis in additional cell lines will need to be carefully evaluated.

It is noteworthy that NINJ1-deficiency does not protect cells from cell death (as assessed by ATP measurements) after treatment with ferroptosis activators. A similar observation has previously been made for pyroptotic cells and it explains why NINJ1 has not been previously identified in whole-genome CRISPR screens that sought to identify drivers of ferroptosis or pyroptosis (Doll et al, 2017; Shi et al, 2015). While it is assumed that pyroptotic cells are killed by the formation of GSDMD pores that disrupt ionic gradients and the mitochondrial membrane potential, it remains unclear how lipid peroxidation kills cells (Feng and Stockwell, 2018). It is possible that the NINJ1-independent opening of plasma membrane channels and/or an accumulation of peroxidized lipids in organelles, such as mitochondria or lysosomes, disturb cellular homeostasis irreparably and thus lead to cell death.

Nevertheless, the fact that NINJ1-deficient cells fail to undergo lysis for hours after ferroptosis induction can have a profound impact on how ferroptotic cells are sensed by the surrounding tissue and the immune system (Sarhan et al, 2018). Our proteomics analysis shows that NINJ1-deficiency strongly reduces the release of cytosolic proteins after ferroptosis induction. Among the proteins released from WT but not from $Ninj1^{-/-}$ cells, we find many potent DAMPs, like heat shock proteins, HMGB1 and histones, some of which were previously reported to be released by ferroptotic cells (Tang et al, 2021). This NINJ1-dependent secretome is similar but not identical to the NINJ1-dependent secretome of pyroptotic cells (Kayagaki et al, 2021). Notably, Hsp90aa1 and Hspa4 are released independently of NINJ1 during pyroptosis, while they require NINJ1-dependent PMR during ferroptosis. This difference is most likely caused by the fact that the GSDMD pores that are formed during pyroptosis are very large and can themselves mediate unconventional protein secretion of certain substrates, while no comparably large pores are formed upstream of NINJ1 during ferroptosis (see above).

Cytosolic proteins released by cell lysis are well known to activate innate immune sensors and thereby drive cytokines production and inflammation. Given that DAMPs are released from ferroptotic cells in a NINJ1-dependent manner, it will be very interesting to investigate the role of NINJ1 in diseases that have been shown to involve ferroptosis, such as acute renal failure caused by ischemia, nephrotoxic drugs or urinary tract obstruction, acute lung injury/acute respiratory distress syndrome (ALI/ARDS) and intestinal bowel disease (IBD). This could be done using NINJ1-deficient animal models or by administrating newly developed anti-NINJ1 antibodies that block oligomerization of the protein and thus PMR. Administration of such neutralizing antibodies has recently been shown to reduce hepatocellular PMR induced with TNFα plus D-galactosamine, concanavalin A, Jo2 anti-Fas agonist antibody or ischemia–reperfusion injury, thereby reducing markers of liver inflammation and necrosis, among them the release of DAMPs (such as HMGB1, alanine aminotransaminase (ALT) and aspartate aminotransferase (AST)) and neutrophil infiltration (Kayagaki et al, 2023).

In summary, our data support a model for ferroptotic cell lysis, in which lipid peroxidation results in increased plasma membrane tension that causes the opening of mechanosensitive ion channels and an ion disbalance that is followed by the activation of NINJ1. Active oligomeric NINJ1 then induces a loss of membrane integrity and, at later time points, complete cell lysis. Moreover, the critical role of NINJ1 in driving PMR and DAMP release suggests that targeting NINJ1 could serve as potential therapeutic approach to limit ferroptosis-associated inflammation.

# Methods

## Animals

All experiments implicating animals (mice C57CL/6, male and female 8–12 weeks) were performed under the guidelines and approval from the cantonal veterinary office of the canton of Vaud (Switzerland), license number VD3257. All mice were bred and housed at a specific-pathogen-free facility at 22 ± 1 C° room temperature, 55 ± 10% humidity and a day/night cycle of 12 h/12 h at the University of Lausanne. *Ninj1*-deficient, *Gsdmd-*, *Gsdme-*, *Nlrp3-*, *Mlkl-* and *Casp8/Ripk3-* mice have been described before (Degen et al, 2023; Chen et al, 2019; Kaiser et al, 2011).

## Plasmids and cloning of NINJ1 mutants for expression in mammalian cells

The plasmids used in this manuscript are all available on Addgene. Doxycycline inducible C-terminal GFP tagged hNINJ1 (hNINJ1-GFP) is Addgene plasmid 208779 and was generated as previously described (Degen et al, 2023). Plasmids for the MSCV experiments are as follows: MSCV-IRES-GFP is available on Addgene (20672); MSCV-WT mouse NINJ1-GFP (Addgene plasmid 208776); MSCV-K45Q-NINJ1-GFP (Addgene plasmid 208777) and MSCV-D53A-NINJ1-GFP (Addgene plasmid 208778) were generated as previously described (Degen et al, 2023).

## Mammalian cell culture and generation

WT, *Ninj1*$^{-/-}$, *Gsdmd*$^{-/-}$, *Gsdme*$^{-/-}$, *Nlrp3*$^{-/-}$, *Mlkl*$^{-/-}$, *Casp8/RipK3*$^{-/-}$ mouse bone marrow-derived macrophages (BMDMs)

from C57BL/6 male and female mice were harvested and differentiated in Dulbecco's modified Eagle medium (DMEM) (Gibco) containing 20% L929L supernatant, as a source of macrophage colony-stimulating factor (M-CSF), 10% heat-inactivated fetal calf serum (FCS) (BioConcept), 10 mM Hepes (BioConcept), 1% penicillin/streptomycin and nonessential amino acids (NEAA, Gibco), and stimulated on days 9 to 10 of differentiation. Human epithelial HeLa cells (clone CCL-2 from ATCC) were cultured in DMEM supplemented with 10% FCS. RAW 264.7 (TIB-71 from ATTC) and NIH/3T3 (CRL-1658 from ATTC) cells were cultured in DMEM supplemented with 10% FCS. MEFs were cultured in DMEM with 10% FCS, 2 mM L-glutamine, and 400 μM sodium pyruvate. WT and Ninj1-deficient MEFs were received from Prof. M. Bertrand (VIB, Ghent, Belgium). Immortalization of macrophages was performed as previously described (Broz et al, 2010). Immortalized macrophages (iBMDMs) were cultured in DMEM containing 10% FCS, 10% M-CSF, 10 mM Hepes, and 1% nonessential amino acids. All cells were grown at 37 °C, 5% CO$_2$. All cell lines are regularly tested in the lab for mycoplasma contamination and are mycoplasma free. Cell lines obtained from ATCC were authenticated by the vendor. The identity of cell lines was frequently checked by their morphological features and did not show any signs of cross-contamination.

## Retroviral transduction of primary BMDMs

Primary BMDMs were transduced as described previously (Broz et al, 2010). Genes encoding WT or mutant mouse NINJ1 were cloned into pMSCV2.2-IRES-GFP. *Ninj1*$^{-/-}$ BMDMs were transduced twice with retroviral particles generated from transfection of Phoenix-Eco packaging cells. The supernatant of the Phoenix Eco cells was added to primary cells via spinfection 48 and 72 h after extraction from bones and cells were cultured as described above. Four days after the first transduction, fully differentiated and transduced BMDMs were seeded for experiments.

## Generation of NINJ1 knockout RAW cells

RAW 264.7 and NIH/3T3 cells were transduced with lentivirus made from Addgene plasmid 83480 targeting NINJ1 (AGAGCT-TACCAAGGCGTCGG) or luciferase as a control (ACAACCGC-GAAAAAGTTGCGCGG). Lentivirus was made by transfecting 3 million 293T cells in a 6-well dish with 1.25 μg of packaging plasmid, 1.25 μg of psPAX2, and 250 ng of VSVG and 8 μL LT-1 transfection reagent (Mirus). Media was changed 24 h post transfection to DMEM + 10% FCS + 1 g/100 mL Bovine Serum Albumin (BSA) and lentivirus was harvested 48 h post transfection and filtered through a 0.45 μm filter. 2 million RAW or 3T3 cells were transduced with 500 μL of each virus by spinning at $500 \times g$ for 1.5 h in one well of a 12-well dish. Next, cells were split out into a T75 flask and selected with 5 μg/mL blasticidin 24 h post transduction. All western blots and experiments were performed on cells at least 7 days post addition of blasticidin.

## Cell lysis assays

A day before stimulation, BMDMs were seeded in 96-well plates at a density of $5 \times 10^4$ per well. To activate ferroptosis CuOOH, RSL3, or ML162 were used at concentrations and time points indicated in

figure legends. To activate ferroptosis, BMDMs were treated in full media for 5 h with CuOOH (1 mM, 400 μM or 20 μM; Sigma-Aldrich), for 6 h or 8 h with RSL3 (25 μM, 5 μM or 0.2 μM; Lubioscience) or for 5 h or 8 h with ML162 (25 μM, 5 μM or 0.2 μM; Sigma-Aldrich). RAW 264.7 and NIH/3T3 cells were seeded in a 96-well plate at a density of $3 \times 10^4$ per well, and the next day treated for 2 h or 5 h with CuOOH (1 mM, 200 μM or 40 μM), for 3 h or 8 h with RSL3 (25 μM, 5 μM or 1 μM) or for 3 h or 8 h with ML162 (25 μM, 5 μM or 1 μM). MEFs were seeded in a 96-well plate at a density of $3 \times 10^3$ per well and the next day treated for 5 h with CuOOH (1 mM 400 μM or 20 μM) or for 8 h with RSL3 (25 μM, 5 μM or 1 μM) or for 8 h with ML162 (25 μM, 5 μM or 1 μM). HeLa cells were seeded in 96-well plates at a density of $2.5 \times 10^4$ cells per well, and the next day treated for 3 h with CuOOH (1 mM) RSL3 (5 μM) or ML162 (5 μM). When indicated ferroptosis inhibitors, see figure legend, Ferrostatin-1 (Fer-1) (25 μM; Sigma-Aldrich) or Liproxtatin-1 (Liprox) (1 μM: Sigma-Aldrich) were added at the same time as ferroptosis activators. To activate canonical inflammasome, WT BMDMs were first primed with $Pam_3CSK_4$ (1 μg/mL; InvivoGen) for 4 h, washed and then stimulated in full media with nigericin (5 μg/mL; Sigma-Aldrich). Cell lysis was quantified by measuring the lactate dehydrogenase (LDH) amount in the cell supernatant using the LDH cytotoxicity kit (Takara, Clontech) according to the manufacturer's instructions and expressed as a percentage of total LDH release. LDH release was normalized to untreated control and 100% lysis control, by adding Triton X-100 to a final concentration of 0.01%: $(LDH_{sample} - LDH_{negative\ control})/(LDH_{100\%\ lysis} - LDH_{negative\ control}) \times 100$.

## Lipid peroxidation measurement

WT and $Ninj1^{-/-}$ BMDMs were seeded in 24-well plates at a density of $2 \times 10^5$ cells per well a day before stimulation. The next day cells were treated with CuOOH (1 mM; Sigma-Aldrich), RSL3 (5 μM; Lubioscience), or ML162 (5 μM; Sigma-Aldrich) for the indicated time (see figure legend). When indicated, Fer-1 (25 μM; Sigma-Aldrich) was added at the same time that ferroptosis activators. After treatment, cells were incubated with C11 BODIPY 581/591 (1 μM; Thermo) for 15 min at 37 °C, washed twice and resuspended in PBS. Cellular fluorescence was evaluated by flow cytometry (CytoFLEX S V4-B2-Y4-R3 Flow Cytometer). FITC-A-channel determined lipid peroxidation and a total of $1 \times 10^4$ or $2 \times 10^4$ cells were analyzed per experiment. The intensity in the negative control sample did not overlap with that in treated cells and a threshold value of less than 2% of oxidized C11 BODIPY 581/591 in control cells was set. The data were analyzed by FlowJo vx10, as a total count of cells, histograms, or as percentage of positive cells for oxidized C11 BODIPY 581/591.

## Confocal microscopy, time-lapse imaging, and image analysis

BMDMs seeded onto glass coverslips were treated with different cell death stimuli, fixed in 4% PFA for 20 min, washed with PBS, permeabilized with 0.05% saponin and blocked with 1% BSA. Then, samples were incubated with an anti-mouse NINJ1 monoclonal antibody (rabbit IgG2b clone 25, Genentech; 1:2000), Alexa Fluor 488- or 568-conjugated anti-rabbit (Life Technologies, 1:500) and Hoechst (1:1000). For morphology analysis of BMDMs, cells were

seeded onto 8-well μ-slides (Ibidi), primed with Pam3CSK4 (1 μg/mL; InvivoGen) for 4 h and treated in opti-MEM with nigericin (5 μg/mL; Sigma-Aldrich), CuOOH (1 mM; Sigma-Aldrich), RSL3 (5 μM; Lubioscience) or ML162 (5 μM; Sigma-Aldrich). Ferrostatin-1 (Fer-1) (25 μM; Sigma-Aldrich) was added at the same time as ferroptosis-inducing treatments. DRAQ7 (1 μM;Bio-Legend) was added to the media to visualize membrane permeabilization and cells were incubated for the indicated time points. Cells were then imaged without fixation, to preserve morphological characteristics. HeLa cells were seeded onto 8-well μ-slides and transiently transfected with plasmids encoding Dox-inducible hNINJ1-GFP, which expression was induced for 16 h (1 μg/mL Dox; Sigma) or $HA^{TMD}$-GFP. Cells were stimulated with CuOOH (1 mM; Sigma-Aldrich) in opti-MEM and DRAQ7 (1 μM) was added to the media to visualize membrane permeabilization. Samples were then imaged with a Zeiss LSM800 confocal laser scanning microscope using a 63x/1.4 NA oil objective. Time-lapse microscopy was performed at 37 °C with controlled humidity and $CO_2$, using a motorized xyz stage with autofocus (Zeiss Definite Focus.2 system), and data were acquired using Zeiss ZEN 2 software. Quantification of single-nuclei DRAQ7 intensities was performed using a maximum projection of the z-stack, by manually segmenting the nucleus area and measuring the intensity density of the selected regions over time. Fluorescence intensity at each time point (Ft) was then normalized to the intensity at the initial time point of the experiment (Fi), i.e., (Ft-Fi)/Fi. The pattern of GFP-tagged proteins distribution overtime on the plasma membrane of HeLa cells was then assessed using Quantitative Analysis of the Spatial-distributions in Images using Mosaic segmentation and Dual parameter Optimization in Histograms (QuASIMoDOH) (Paparelli et al, 2016), at the basal plane of cells. The distribution inhomogeneity at each time point (Dt) was normalized to the distribution inhomogeneity at the initial time point of the experiment (Di), i.e., Dt-Di. Time t = 0 was defined when DRAQ7 (Ft-Fi)/Fi > 1, which corresponds to a significant increase in DRAQ7 nuclear fluorescence intensity and onset of plasma membrane permeabilization. All microscopy datasets were analyzed and processed using Fiji software.

## Crosslinking assays

WT BMDMs were seeded in 24-well plates at a density of $2 \times 10^5$ cells per well a day before stimulation. The next day, cells were treated in full medium with CuOOH (1 mM; Sigma-Aldrich) for 3 h, RSL3 (5 μM; Lubioscience) for 5 h, or ML162 (5 μM; Sigma-Aldrich) for 4 h to induce ferroptosis. To induce pyroptosis, BMDMs were first primed in full medium with $Pam_3CSK_4$ (1 μg/mL; InvivoGen) for 4 h followed by stimulation with nigericin (5 μg/mL; Sigma-Aldrich) for 1.5 h in PBS. When indicated, Fer-1 (25 μM; Sigma-Aldrich) was added at the same time as ferroptosis activators and maintained throughout the entire incubation. For nigericin-treated cells, Fer-1 was added 1 h prior nigericin stimulation. Next, the crosslinker $BS^3$ (bis(sulfosuccinimidyl)suberate) was added according to the manufacturer's instructions. In brief, $BS^3$ was added to the media (3 mM; Thermo Fisher) and incubated for 15 min at room temperature. Next, a solution of 20 mM Tris pH 7.5 was added to stop the reaction and incubated for 15 min at room temperature. Cell supernatants were collected, and proteins were precipitated and combined with cell lysates for western blotting analysis.

## Immunoblotting

For western blotting analysis, cells were lysed in 66 mM Tris-HCl pH 7.4, 2% SDS, 10 mM DTT, and NuPage LDS sample buffer (Thermo Fisher). When mentioned, supernatants were precipitated by methanol and chloroform, using standard methods, and pooled with the cell lysates. Proteins were separated on gradient precast gels 4–20% (Millipore) and transferred onto nitrocellulose membrane using Transblot Turbo (Bio-Rad). The antibodies used were: anti-mouse NINJ1 monoclonal antibody (rabbit IgG2b clone 25; a kind gift from Genentech; 1:8000), mouse anti-tubulin (ab40742; Abcam; 1:2000), mouse anti-GAPDH (365062; Santa Cruz, 1:3000) and anti-vinculin (ab91459; Ab91459, 1:1000). Primary antibodies were detected with horseradish peroxidase (HRP)-conjugated goat anti-rabbit (4030-05; Southern Biotech; 1:5000), HRP-conjugated goat anti-mouse (1034-05; Southern Biotech; 1:5000 or 12-349; MilliporeSigma; 1:2000) secondary antibodies.

## Cell viability assay

A day before stimulation, WT and $Ninj1^{-/-}$ BMDMs were seeded in 96-well plates at a density of $5 \times 10^4$ per well in complete media. The next day, cells were treated with CuOOH (1 mM; Sigma-Aldrich), and after the indicated time (see figure legend) supernatant was removed and 50 µl of CellTiter-Glo suspension (Promega) was added on top of the cells and incubated for 5 min at RT. The absorbance of each culture well was measured with a microplate reader (Spectramax i3; Molecular devices). A 100% cell death control was obtained by adding Triton X-100 to a final concentration of 0.01% and the percentage of ATP was calculated as follows: $(ATP_{sample} - ATP_{100\% death})/(PI_{negative control} - PI_{100\% perm.}) \times 100$.

## Calcium fluxes measurement

A day before stimulation, WT and $Ninj1^{-/-}$ BMDMs were seeded in 96-well plates at a density of $5 \times 10^4$ per well in serum-free media. The next day, cells were loaded with Fluo-8 Ca flux assay reagent (Abcam) following the manufacturer's instructions. In brief, 100 µl of 1X reagent were added on top of the cells (100 µl of serum-free media) and were incubated for 30 min at 37 °C and 30 min at RT. After incubation, cells were treated with CuOOH (1 mM; Sigma-Aldrich) in the presence or absence of Ferrostatine-1 (Fer-1) (25 µM; Sigma-Aldrich) and cellular fluorescence was measured over time, reading every 8 min, using a fluorescence plate reader (Cytation 5; Biotek). Fluorescence intensity at each time point was then normalized to the intensity at the same time point in untreated (UT) cells: (Fluorescence sample $_{time = x}$ – Fluorescence UT $_{time = x}$).

## Cell permeabilization assays (loss of membrane integrity)

Propidium iodide (PI) uptake quantification assay: PI was added to the media (12.5 µg/mL; Thermo Fisher Scientific) simultaneously with the specific treatment, and its influx was measured over time, reading every 10 min, using a fluorescence plate reader (Cytation 5; Biotek). A 100% permeabilization control was obtained by adding Triton X-100 to a final concentration of 0.05%, and the percentage of PI uptake was calculated as follows: $(PI_{sample} - PI_{negative control})/$ $(PI_{100\% perm.} - PI_{negative control}) \times 100$. PI images were taken using an imaging plate reader (Cytation 5; Biotek).

## Dextran dye release assay

Before seeding, WT and $Ninj1^{-/-}$ BMDMs were loaded with dextran dye conjugate Alexa Fluor™ 488; 3,000 MW (50 µg/mL, Thermo Fisher Scientific) using Amaxa® Mouse Macrophage Nucleofector® Kit (Lonza). WT and $Ninj1^{-/-}$ BMDMs were loaded according to the manufacturer's instructions. In brief, 1 million BMDMs were resuspended in 100 µl Supplemented Nucleofector® Solution with dextran dye and electroporated (Nucleofector® Program Y-001). After loading, the cells were seeded, see above, and 5 h later were treated with CuOOH (1 mM; Sigma-Aldrich). Images were taken for 8 h, reading every 20 min, at 10× magnification on a IncuCyte SX1 (Sartorious). Two or three images per well were captured and averaged. Data was collected as dextran count per image and normalized as the maximum dextran count per treatment corresponding to 100% dextran count.

## SILAC labeling of iBMDMs and secretome sampling

WT and $Ninj1^{-/-}$ iBMDMs were cultured in SILAC media (containing light or heavy amino acid isotopes), supplemented with 10% dialyzed FCS and 10% MCSF for 1.5 weeks (5 passages). The day before CuOOH treatment, 10 million cells were seeded per condition in a 10 cm dish. Cells were treated with plain SILAC media, i.e., not supplemented with labeled amino acids, FBS or MCSF, containing CuOOH (1 mM; Sigma-Aldrich) for 2 h. Next, supernatants were collected and spun twice at $500 \times g$, and precipitated by methanol and chloroform, using standard methods and resuspended in FASP buffer (4% SDS, 10 mM DTT, 100 mM Tris pH 7.5), heated to 95 °C. The subsequent steps (digestion, fractionation, and LC-MS were conducted at the PAF of the University of Lausanne.

## Protein digestion and fractionation

Protein concentration was determined using the tryptophane fluorescence method (Wiśniewski and Gaugaz, 2015). H and L samples were mixed equimolarly (total 60 µg), and were digested following the SP3 method (Hughes et al, 2019) using magnetic Sera-Mag Speedbeads (Cytiva 45152105050250, 50 mg/ml). Briefly, proteins were alkylated with 32 mM (final) iodoacetamide for 45 min at RT in the dark. Beads were added at a ratio 10:1 (w:w) to amount of material, and proteins were precipitated on beads with ethanol (final concentration: 60%). After three washes with 80% ethanol, beads were digested in 50 µl of 100 mM ammonium bicarbonate with 1.0 µg of trypsin (Promega #V5073). After 2 h of incubation at 37 °C, the same amount of trypsin was added to the samples for an additional 1 h of incubation. Supernatants were then recovered, supplemented with two sample volumes of isopropanol containing 1% TFA and desalted on a strong cation exchange (SCX) plate (Oasis MCX; Waters Corp., Milford, MA) by centrifugation. After washing with isopropanol/1%TFA, peptides were eluted in 200 µl of 80% MeCN, 19% water, 1% (v/v) ammonia, and dried by centrifugal evaporation. Digests were redissolved and fractionated in 6 fractions using the Pierce High pH Reversed-Phase Peptide Fractionation Kit (Thermo Fisher Scientific). The

fractions collected were in 7.5, 10, 12.5, 15, 20, and 50% acetonitrile in 0.1% triethylamine (~pH 10). Dried bRP fractions were redissolved in 50 µl 2% acetonitrile with 0.5% TFA, and 5 µl were injected for LC-MS/MS analyses.

## Liquid chromatography-coupled mass spectrometry (LC-MS) analysis

Data-dependent LC-MS/MS analyses of samples were carried out on a Fusion Tribrid Orbitrap mass spectrometer (Thermo Fisher Scientific) connected through a nano-electrospray ion source to an Ultimate 3000 RSLCnano HPLC system (Dionex), via a FAIMS interface. Peptides were separated on a reversed-phase custom packed 45 cm C18 column (75 µm ID, 100 Å, Reprosil Pur 1.9 µm particles, Dr. Maisch, Germany) with a 4–90% acetonitrile gradient in 0.1% formic acid (total time 140 min). The used MS acquisition method cycled through three compensation voltages ($-40$, $-50$, $-60$ V) to acquire full MS survey scans at 120,000 resolution. A data-dependent acquisition method controlled by Xcalibur software (Thermo Fisher Scientific) was set up that optimized the number of precursors selected ("top speed") of charge 2+ to 5+ from each survey scan, while maintaining a fixed scan cycle of 1.0 s per FAIMS CV. Peptides were fragmented by higher energy collision dissociation (HCD) with a normalized energy of 32%. The precursor isolation window used was 1.6Th, and the MS2 scans were done in the ion trap. The m/z of fragmented precursors was then dynamically excluded from selection during 60 s.

## Mass spectrometry data processing and analysis

Data files were analyzed with MaxQuant 2.1.4.0. (Cox and Mann, 2008) incorporating the Andromeda search engine (Cox et al, 2011). Cysteine carbamidomethylation was selected as fixed modification while methionine oxidation and protein N-terminal acetylation were specified as variable modifications. SILAC heavy labeling was specified as Lys+8 and Arg+10. The sequence databases used for searching were the mouse (*Mus musculus*) reference proteome based on the UniProt database (RefProt_-Mus_musculus_20230301.fasta, from www.uniprot.org, version of January 2023, containing 55,309 sequences), and a "contaminant" database containing the most usual environmental contaminants and enzymes used for digestion (e.g., keratins, trypsin). Mass tolerance was 4.5 ppm on precursors (after recalibration) and 20 ppm on MS/MS fragments. Both peptide and protein identifications were filtered at 1% FDR relative to hits against a decoy database built by reversing protein sequences. Filtering and processing of MaxQuant outputs were done with the Perseus software package (version 1.6.15.0) (Tyanova et al, 2016). Contaminant proteins were removed, and SILAC ratios were $\log_2$-transformed.

To not skew enrichment distributions, raw enrichment values (heavy-to-light-ratios) were taken for further analysis, and to avoid false positives, missing values were excluded. NINJ1-dependent protein release was characterized by presence of a given protein in the wildtype secretome upon treatment (either with respect to untreated wildtype cells, H/L-ratio > 0.5, or with respect to treated *Ninj1*$^{-/-}$ iBMDMs, H/L-ratio > 1). Proteins that were identified in *Ninj1*$^{-/-}$ iBMDMs upon treatment (ratio > 0.5) were deemed to be NINJ1-independently released. For the differential enrichment of GO-terms in Cellular Localization or Biological Process, Chi-square test with the base assumption of equal distribution was performed.

## Treatment of recipient cells with ferroptotic secretomes

WT and *Ninj1*$^{-/-}$ iBMDMs were cultured as described above. Cells were seeded at a density of $3 \times 10^4$ cells per well of a 96-well plate one day before the experiment. The following day, the medium was changed for either Optimem, Optimem containing CuOOH (1 mM; Sigma-Aldrich), or Optimem containing 1 mM CuOOH and 25 µM Ferrostatin-1. After treatment for 2 h, the supernatant containing ferroptosis activator was removed and replaced with fresh Optimem. After 1 h the medium was collected (donor medium) and transferred onto naïve WT iBMDMs which had been seeded in parallel to the donor plate at the same cell density. LDH release assay was performed on both the treatment media and the donor media to confirm ferroptosis induction. The recipient plate was incubated at 37 °C for 16 h and subsequently, the supernatant was collected and tested for LDH release. The cells were harvested, RNA was extracted using the Qiagen RNeasy kit (Cat. Nr. 74004), following the manufacturer's instructions.

## Reverse transcription (RT) and qPCR

Extracted and DNAse-I treated RNA was adjusted for concentration, and RT was performed using AMV-RT. First, 4 µl RNA were mixed with 1 µl random primer and incubated for 5 min at 70 °C. Subsequently, the RT mix, containing AMV buffer, dNTPs, MgCl$_2$, RNAse-inhibitor, and AMV-RT was added and incubated at 42 °C for 1 h, followed by 99 °C for 5 min. The final product was diluted 1:10 and used for qPCR, using SYBR Green 2x Master Mix and the following primers: 18S$_{fwd}$: GTAACCCGTTGAACCCCATT, 18S$_{rev}$: CCATCCAATCGGTAGTAGCG, TNF$_{fwd}$: AAGGGGATTATGGCTCAGGG, TNFa$_{rev}$: AGGCTCCAGTGAATTCGGAA, IL1ß$_{fwd}$: ACTCATTGTGGCTGTGGAGA, IL1ß$_{rev}$: TTGTTCATCTCGGAGCCTGT. qPCR was conducted on a Roche Light Cycler 480, and analysis of C$_t$-values was performed in the manufacturer's software. To pool data, C$_t$-values were normalized to the 18S control fold changes in expression level compared to untreated WT iBMDMs were calculated.

## Experiment study design

There was no randomization for these experiments. This study is not a randomized control trial and randomization is not conventionally used in in vitro/in cellulo studies such as this one. All groups of experiments were performed using the same protocols and experimental conditions. When using primary mouse bone marrow-derived macrophages, cells from at least 2 animals per genotype were used for reproducibility. Blinding Assays used objective quantification methods that are not susceptible to bias, so samples were not blinded. The only data excluded for analysis were outlier values, due to errors during the procedure or on the instruments performed for the experiment (e.g., no signal in one well).

## Data analysis and statistics

Data analysis was performed using GraphPad Prism v9 and Microsoft Excel. All microscopy datasets were analyzed and

processed using Fiji software. No statistical method was used to determine sample sizes and no sample size calculation was performed. Statistical significances were assessed using Student's two-tailed *t*-test (for comparison of two groups) or one-way analysis of variance (ANOVA) for multiple comparisons tests when comparing repeated measures over time or two-way ANOVA for multiple comparisons tests when comparing three or more groups. Significances are referred as *, **, ***, or ****, for *P*-values < 0.05, <0.01, <0.001, or <0.0001, respectively.

## Data availability

The mass spectrometry proteomics data have been deposited to the ProteomeXchange Consortium (Deutsch et al, 2023) via the PRIDE partner repository (Perez-Riverol et al, 2022) with the dataset identifier: PXD047622. The Source data for microscopy images of the main figures has been deposited to the BioStudies repository, accession number: S-BSST1254.

## Peer review information

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

## Acknowledgements

This work is supported by ERC-CoG 770988 (InflamCellDeath) and SNF Project funding (310030B_198005, 310030B_192523) to PB, EMBO postdoctoral fellowships ALTF 27-2022 to EH and ALTF 566-2022 to PW. We would like to thank Prof. M. Bertrand (VIB, Gent, Belgium) for WT and *Ninj1*-deficient MEFs, the UNIL cellular Imaging Facility (CIF), the protein analysis facility (PAF) and the UNIL animal facility for their support.

## Author contributions

**Saray Ramos**: Conceptualization; Data curation; Software; Formal analysis; Validation; Investigation; Methodology; Writing—original draft; Project administration; Writing—review and editing. **Ella Hartenian**: Data curation; Software; Formal analysis; Validation; Investigation; Methodology; Writing—review and editing. **José Carlos Santos**: Data curation; Software; Formal analysis; Methodology; Writing—review and editing. **Philipp Walch**: Data curation; Software; Formal analysis; Writing—review and editing. **Petr Broz**: Conceptualization; Resources; Supervision; Funding acquisition; Validation; Investigation; Writing—original draft; Project administration; Writing—review and editing.

## Disclosure and competing interests statement

The authors declare no competing interests.

# Expanded View Figures

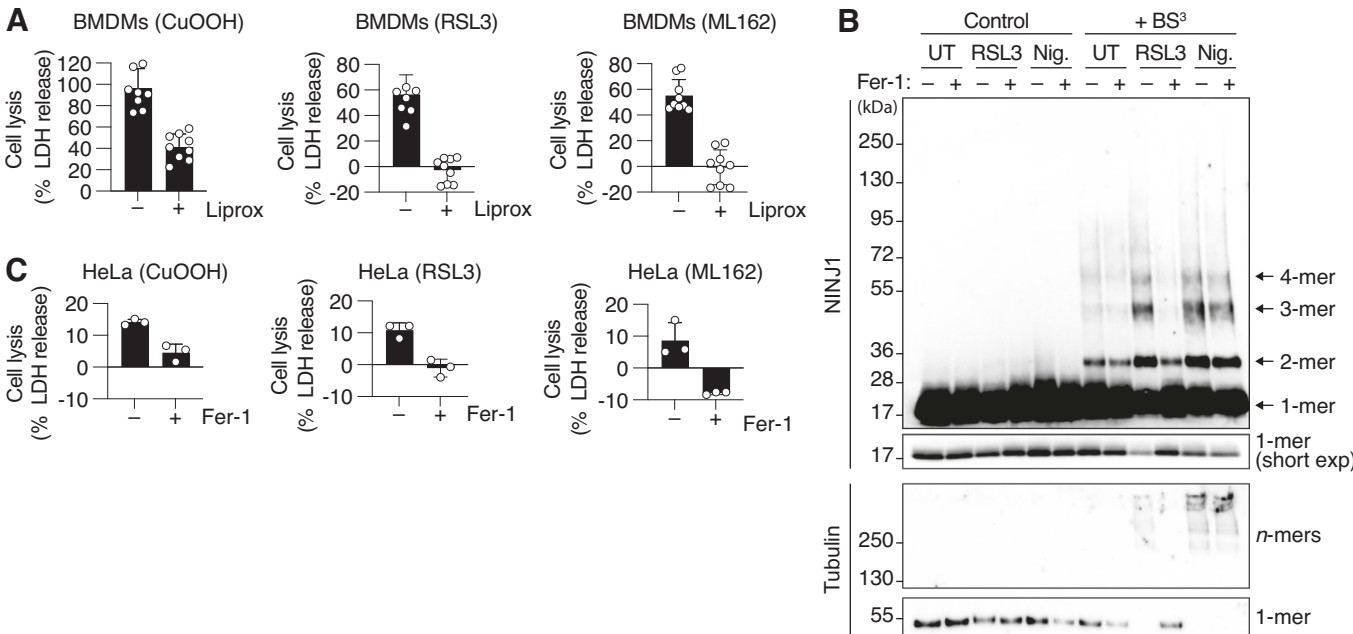

**Figure EV1.  Ferroptosis inhibitors specifically block ferroptosis in BMDMs.**

(**A**) LDH release in WT BMDMs treatment with 1 mM CuOOH for 5 h, 5 μM RSL3 for 6 h, or 5 μM ML162 for 5 h. When indicated, 1 μM Liprox was added simultaneously with CuOOH, RSL3, or ML162. Liprox = Liproxstatin-1. (**B**) Western blot analysis of endogenous NINJ1 in WT BMDMs treated with 5 μg/mL nigericin for 1.5 h or 5 μM RSL3 for 4 h followed by treatment with the membrane-impermeable crosslinker BS$^3$. Mixed supernatant and cell extracts were analyzed. FL, full length; Short exp = short exposure; 1-mer = monomer; 2-mer = dimer; 3-mer = trimer; 4-mer = tetramer and n-mer, higher-order oligomers. (**C**) LDH release in HeLa cells after treatment with 1 mM CuOOH for 3 h, 5 μM RSL3 for 3 h, or 5 μM ML162 for 3 h. When indicated, 25 μM Fer-1 was added simultaneously with ferroptosis and pyroptosis activators (**B**, **C**). Data information: All graphs show the mean ± SD. Data are pooled from three independent experiments (**A**) or representative of two (**B**, **C**) independent experiments performed in triplicate.

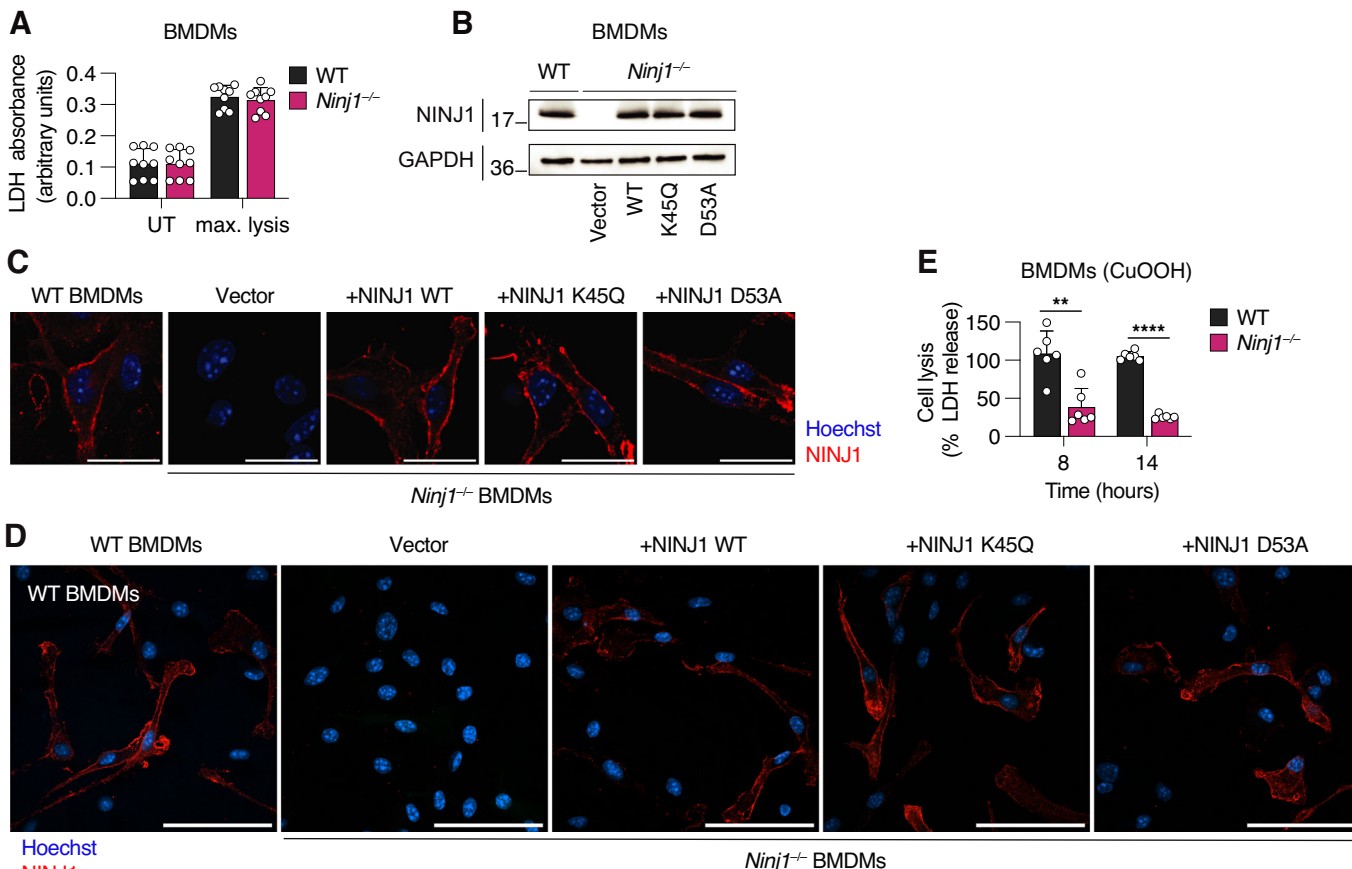

**Figure EV2. NINJ1 filament formation is essential to induce cell lysis in ferroptosis.**

(**A**) LDH absorbance in WT and *Ninj1*$^{-/-}$ BMDMs untreated (UT) or lysed with Triton X-100 to a final concentration of 0.01% (max. lysis). (**B**) Western blot of NINJ1 expression in WT or *Ninj1*$^{-/-}$ BMDMs transduced with a retroviral vector expressing WT mNINJ1 or different mNINJ1 mutants. Transduction with a GFP expressing vector was used as a control. Cell extracts were analyzed. GAPDH is a loading control. (**C, D**) Immunofluorescence confocal microscopy of NINJ1 (red) in WT or *Ninj1*$^{-/-}$ BMDMs complemented with WT or different NINJ1 mutants upon retroviral transduction. Scale bars: 20 μm (**C**) and 60 μm (**D**). (**E**) LDH release in WT and *Ninj1*$^{-/-}$ BMDMs upon treatment with 1 mM CuOOH for 8 or 14 h. Data information: All graphs show the mean ± SD. Data are pooled from three independent experiments performed in triplicate (**A, E**) or representative of two different experiments (**B, C, D**). Statistical analysis was done using Student's unpaired two-sided t-test. **** $P < 0.0001$, ** $< 0.01$.

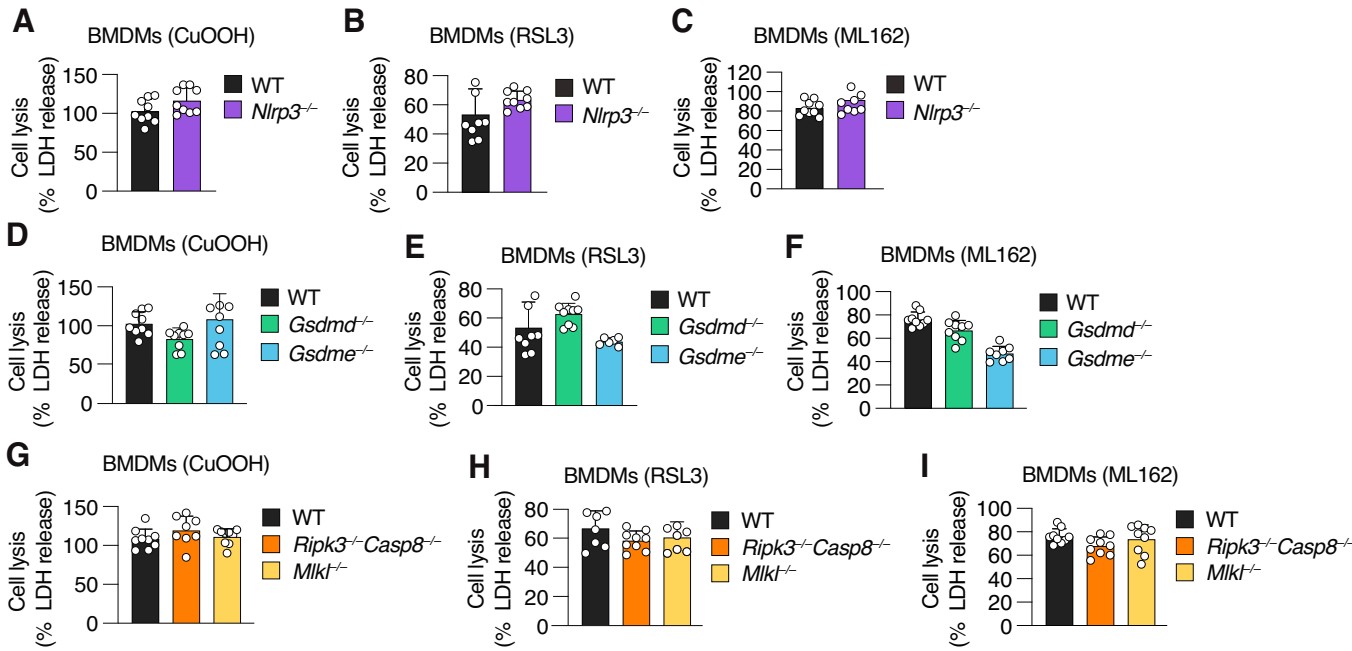

**Figure EV3. NINJ1 is the only driver of cell lysis during ferroptosis.**

(A–I) LDH release in WT and *Nlrp3*⁻/⁻ BMDMs treated with 1 mM CuOOH for 5 h (A), 5 µM RSL3 for 6 h (B), or 5 µM ML162 for 5 h (C), WT, *Gsdmd*⁻/⁻, and *Gsdme*⁻/⁻ BMDMs treated with 1 mM CuOOH for 5 h (D), 5 µM RSL3 for 6 h (E) or 5 µM ML162 for 5 h (F) and WT, *Ripk3*⁻/⁻/*Casp8*⁻/⁻, and *Mlkl*⁻/⁻ treated with 1 mM CuOOH for 5 h (G), 5 µM RSL3 for 6 h (H), or 5 µM ML162 for 5 h (I). Data information: All graphs show the mean ± SD. Data are pooled from three independent experiments performed in triplicate (A–I). Statistical analysis was done using Student's unpaired two-sided t-test (A–C) or one-way ANOVA (D–I).

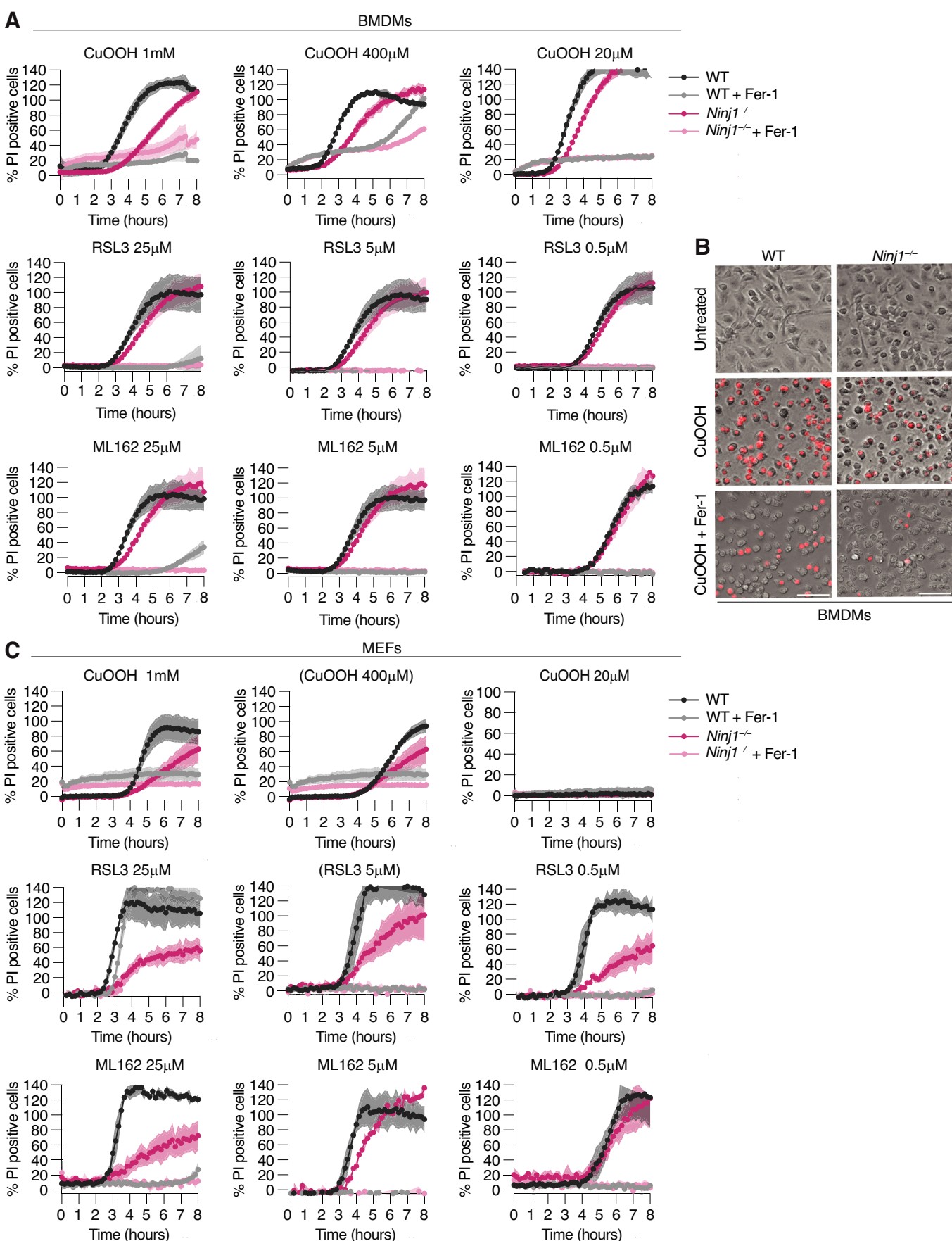

◄ **Figure EV4. NINJ1 controls plasma membrane permeabilization during ferroptosis in macrophages and fibroblasts.**

(A) Percentage of propidium iodide (PI, Mw = 668 Da) uptake in WT and *Ninj1*[−/−] BMDMs over time (0–8 h) after treatment with different concentrations of CuOOH, RSL3, or ML162. (B) Representative images showing PI uptake (red) in WT and *Ninj1*[−/−] BMDMs left untreated or after treatment with 1 mM CuOOH for 5 h. Scale bar: 200 μm. (C) Percentage of propidium iodide (PI, Mw = 668 Da) uptake in WT and *Ninj1*[−/−] MEFs over time (0–8 h) after treatment with different concentrations of CuOOH, RSL3, or ML162. When indicated, 25 μM Fer-1 was added simultaneously with ferroptosis activators (A–C). Data information: All graphs show the mean ± SD. Data are representative of three (A, C) or two (B) independent experiments performed in triplicate.

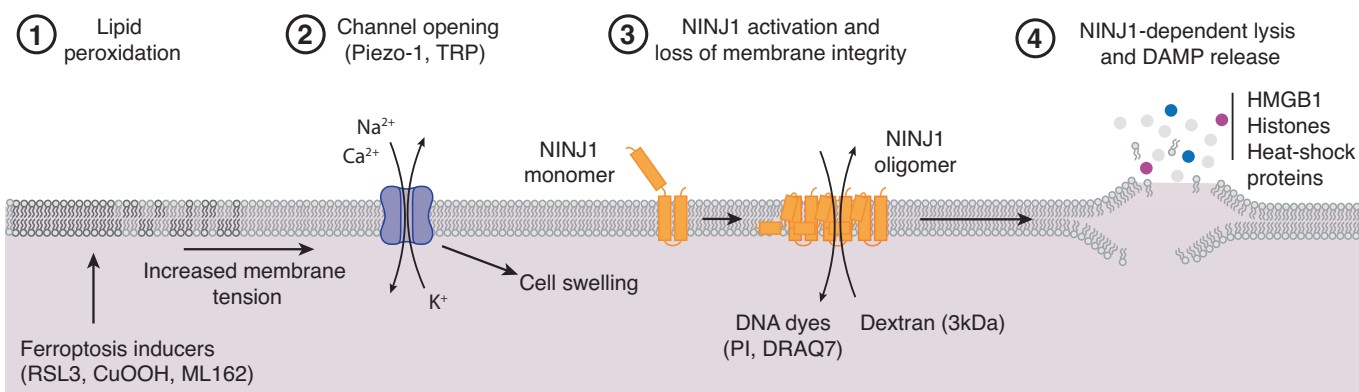

**Figure EV5.  Model for ferroptosis-associated cell lysis.**

(1) Ferroptosis inducers promote the formation of lipid peroxides at the plasma membrane which cause and increase in membrane tension. (2) Mechanosensitive ion channels (Piezo-1 and TRP channels) open is response to increased membrane tension, resulting in the entry of calcium and sodium ions and the release of potassium ions. These non-selective ion fluxes cause an increase in the cell volume and swelling (3) Following this step, NINJ1 is activated, possibly because of ion disbalance, swelling or a yet unrecognized signal, and oligomerizes to form amphipathic filaments. Initially these filaments cause only small lesions that allow the entry and exit of small molecules, such as DNA-binding dyes. (4) Eventually NINJ1 lesions cause complete membrane rupture and cell lysis. As a consequence of NINJ1-dependent PMR, ferroptotic cells start to leak intracellular content among them many DAMPs that can drive inflammation.

