## [Peer Review File · The EMBO Journal]

NINJ1 induces plasma membrane rupture and release of damage-associated molecular pattern molecules during ferroptosis

Saray Ramos, Ella Hartenian, José Santos, Philipp Walch, and Petr Broz

Corresponding author(s): Petr Broz (petr.broz@unil.ch)

Review Timeline:

Submission Date:	18th Jul 23
Editorial Decision:	18th Aug 23
Revision Received:	14th Dec 23
Editorial Decision:	16th Jan 24
Revision Received:	26th Jan 24
Accepted:	30th Jan 24

Editor: Ioannis Papaioannou

Transaction Report:

Dear Prof. Broz,

Thank you for submitting your manuscript for consideration by the EMBO Journal. It has now been seen by three experts in the field, and we have received the full set of their reports, which are included below.

As you will see, all referees acknowledge that the findings are significant, interesting, and timely, and they think that the data are convincing and largely support the conclusions of the study. However, they also raise a few concerns, including the cell type and stimulus specificity of the NINJ1 requirement for ferroptosis-induced plasma membrane rupture that is suggested by the available data. I would like to note that this is also relevant to the contradictory results recently reported by Hirata et al. in April 2023 (PMID: 36898371), and we agree with the referees that it should be characterized further. The referees also provide a number of additional suggestions for the improvement of the study and the manuscript, which should be addressed.

Given the referees' positive comments and recommendations, I would like to invite you to submit a revised version of your manuscript, addressing the comments of all three reviewers. I should add that it is EMBO Journal policy to allow only a single round of major revision, and acceptance of your manuscript will therefore depend on the completeness of your responses in this revised version. If you have any questions or comments, we can also discuss the revisions in a video chat, if you like.

We generally allow three months as standard revision time (17th November 2023). As a matter of policy, competing manuscripts published during this period will not negatively impact our assessment of the conceptual advance presented by your study. However, we request that you contact us as soon as possible upon publication of any related work, to discuss how to proceed. Should you foresee a problem in meeting this three-month deadline, please let us know in advance and we may be able to grant an extension.

Thank you for the opportunity to consider your work for publication in the EMBO Journal. I look forward to your revision.

Yours sincerely,

Instructions for preparing your revised manuscript

1. When you are ready to submit the revision, please upload:

- A Word file of the manuscript text (including legends of main Figures, EV Figures and Tables). Please make sure that changes are highlighted (or "tracked") to be clearly visible.

- Individual production-quality figure files (one file per figure). When assembling your figures, please refer to our figure preparation guidelines in order to ensure proper formatting and readability in print as well as on screen:

If the data shown in a figure are obtained from n {less than or equal to} 2, please use scatter plots showing the individual data points.

- i. the name of the statistical test used to generate error bars and P values
- ii. the number (n) of independent experiments (please specify technical or biological replicates) underlying each data point (discussion of statistical methodology can be reported in the Materials and Methods section, but figure legends should contain a basic description of n , P , and the test applied)
- iii. the nature of the bars and error bars (s.d., s.e.m.).

- A point-by-point response to the referees' comments, with a detailed description of the changes made (as a word file). All referees' concerns must be fully addressed and their suggestions taken on board. When preparing your letter of response to the referees' comments, please bear in mind that this will form part of the Review Process File and will therefore be available online to the community. Please note that you have the possibility to opt out of the transparent process at any stage prior to publication by letting the editorial office know (contact@embojournal.org); if you do opt out, the Review Process File link will point to the following statement: "No Review Process File is available with this article, as the authors have chosen not to make the review process public in this case.". For more details on our Transparent Editorial Process, please visit our website:

<https://www.embopress.org/page/journal/14602075/authorguide#transparentprocess>

- Expanded View (EV) files (replacing Supplementary Information) that are collapsible/expandable online. A maximum of 5 EV Figures can be typeset. EV Figures should be cited as "Figure EV1, Figure EV2" etc. in the text, and their respective legends should be included in the manuscript file after the legends of regular figures. See detailed instructions regarding Expanded View files here:

- For the figures that you do NOT wish to display as Expanded View figures, they should be bundled together with their legends in a single PDF file called "Appendix", which should start with a short Table of Contents (including page numbers). Appendix figures should be referred to in the main text as: "Appendix Figure S1, Appendix Figure S2" etc. Please see detailed instructions here: <https://www.embopress.org/page/journal/14602075/authorguide#expandedview>

- A complete author checklist, which you can download from our author guidelines (<https://www.embopress.org/page/journal/14602075/authorguide>). Please note that the checklist will also be part of the Review Process File.

2. Please note that no statistics should be calculated if $n=2$.

3. Before submitting your revision, primary datasets (and computer code, where appropriate) produced in this study need to be deposited in appropriate public databases (see <https://www.embopress.org/page/journal/14602075/authorguide#dataavailability>). Specifically, we would kindly ask you to provide public access to the following datasets/data:

- Mass spectrometry data.

The accession numbers and database should be listed in a formal "Data availability" section (placed after Materials and Methods) that follows the model below (see also

<https://www.embopress.org/page/journal/14602075/authorguide#dataavailability>):

Data availability

- RNA-seq data: Gene Expression Omnibus GSE46843 (<https://www.ncbi.nlm.nih.gov/geo/query/acc.cgi?acc=GSE46843>)
- [data type]: [name of the resource] [accession number/identifier/doi] ([URL or identifiers.org/DATABASE:ACCESSION])

*** Note: all links should resolve to a page where the data can be accessed. ***

*** Note: the Data Availability Section is restricted to new primary data that are part of this study. ***

4. Please check that the title and the abstract of the manuscript are brief, yet explicit, even to non-specialists. The length of the title should not exceed 100 characters (including spaces), and the abstract should be a single paragraph not exceeding 175 words.

5. Please also note our reference format: <https://www.embopress.org/page/journal/14602075/authorguide#referencesformat>.

7. Please remember: digital image enhancement is acceptable practice, as long as it accurately represents the original data and conforms to community standards. If a figure has been subjected to significant electronic manipulation, this must be noted in the figure legend or in the "Materials and Methods" section. The editors reserve the right to request original versions of figures and the original images that were used to assemble the figure.

8. Our journal encourages inclusion of data citations in the reference list to directly cite datasets that were obtained from public databases. Data citations in the article text are distinct from normal bibliographical citations and should directly link to the database records from which the data can be accessed. In the main text, data citations are formatted as follows: "Data ref: Smith et al, 2001" or "Data ref: NCBI Sequence Read Archive PRJNA342805, 2017". In the Reference list, data citations must be labeled with "[DATASET]". A data reference must provide the database name, accession number/identifiers, and a resolvable link to the landing page from which the data can be accessed at the end of the reference. Further instructions are available at: <https://www.embopress.org/page/journal/14602075/authorguide#referencesformat>.

9. We request authors to consider both actual and perceived competing interests. Please review our policy (<https://www.embopress.org/page/journal/14602075/authorguide#conflictsofinterest>) and update your competing interests statement if necessary. Please name this section 'Disclosure and competing interests statement' and place it after the Acknowledgements section.

10. Please note that all corresponding authors are required to provide an ORCID ID upon submission of a revised manuscript (<https://orcid.org/>). Please find instructions on how to link your ORCID ID to your account in our manuscript tracking system in our Author guidelines (<https://www.embopress.org/page/journal/14602075/authorguide#authorshipguidelines>).

11. We use CRediT to specify the contributions of each author in the journal submission system. CRediT replaces the author contribution section, which should be removed from the manuscript. Please use the free text box to provide more detailed descriptions. See also guide to authors: <https://www.embopress.org/page/journal/14602075/authorguide#authorshipguidelines>.

13. We would also welcome the submission of cover suggestions or motifs to be used by our Graphics Illustrator in designing a cover.

14. Please use the link below to submit your revision:
<https://emboj.msubmit.net/cgi-bin/main.plex>

Referee #1:

Ramos and colleagues study investigate the role of NINJ1 in the lysis of cells during ferroptosis. Using BMDMs as their main model, they demonstrate a role for NINJ1 in regulated cell lysis following two different ferroptotic stimuli, in line, NINJ1 is found to oligomerize. As a functional output of this, the authors demonstrate reduction of DAMP release from NINJ1 deficient BMDMs, leading them to speculate that NINJ1 regulates the inflammatory potential of ferroptosis. Overall, the authors present an interesting, timely study and the data largely support their conclusions. Nonetheless, a couple of points relating to the generality of their findings and the inflammatory consequences of NINJ1 expression during ferroptosis.

- The authors have primarily used BMDMs in the present study to study the role on NINJ1 in plasma membrane rupture thus a general requirement for NINJ1 in ferroptosis induced plasma membrane rupture is unclear. To address this, I would suggest investigating additional cell types (generating NINJ1 deficient cells via CRISPR/Cas9), measuring impact on ferroptosis induced cell lysis by LDH release.

With respect to the above point the authors demonstrate that plasma membrane lysis induced by CuOOH-treatment (but not RSL3) is NINJ1 dependent (suggestive of cell type and stimulus specificity), it is important to determine whether CuOOH is causing ferroptosis in these cells (i.e. can ferrostatin suppress).

- The authors propose that NINJ1 regulates the inflammatory potential of ferroptosis (via promoting DAMP release), based on differential release of DAMPs from dying cells, though this is never formally test. This hypothesis could be directly investigated by transferring supt. from NINJ1 proficient/deficient cells triggered to undergo ferroptosis to recipient BMDMs and monitoring impact of inflammatory signaling pathways/transcriptional outputs.

Referee #2:

In this study, Ramos et al. show evidence that the executor of plasma membrane rupture (PMR), NINJ1, is activated during ferroptosis and mediates the initial loss of membrane integrity, which is different from e.g., pyroptosis where gasdermin pores precedes NINJ1 activation and PMR. The significance of NINJ1 PMR in ferroptosis is suggested from it mediating the release of DAMPs which can drive tissue inflammation. The data are convincing, and the manuscript is well written and easy to read/follow. Some specific concerns/comments that should be addressed:

1. Two different inducers of ferroptosis are used: RSL3, an inhibitor of the antioxidant enzyme GPX4 which will inhibit the reduction of lipid peroxides (but probably incompletely), and Cumene hydroperoxide (CuOOH) which is a strong inducer of lipid peroxidation. From the results throughout the manuscript, it seems like CuOOH is a stronger inducer of ferroptosis than RSL3,

and whereas Ferrostatin inhibits both inducers equally well, deletion of NINJ1 seems to more strongly impact CuOOH-induced ferroptosis (all steps). This is most prominent in RAW-cells (Figure EV2). Could some of this be explained by suboptimal doses used for RSL3? How consistent is this in different cell types? Figure 5 shows differences in kinetics of PI influx, but it would be nice to see dose-responses of CuOOH and RSL3 in the cells used (BMDMs, RAWs and HeLa cells) with regards to cell death (Draq7 and LDH), and including Fer-1 and NINJ1-deficiency, to better understand these observations.

2. Figure 4 shows reduced efficacy of Fer-1 in inhibiting lipid peroxidation starting 1h post ferroptosis induction (CuOOH), and at 3h there is no difference in Fer-1 treated and untreated cells. Is this similar for RSL3? Is inhibition of cell death similarly reduced over time (effect of Fer-1) - or for other functions? E.g., in Figure 2C, Fer-1 still prevents Draq7 influx in response to CuOOH or RSL3 at 5h? How do the authors explain these data when, according to Figure 4, lipid peroxidation is at the same level in Fer-1 treated or untreated cells?

3. Figure 5 is central for the claim that NINJ1 is required for initial membrane damage/permeability in ferroptosis. Given the differences seen in BMDMs and RAW cells, it would be nice to see Figure 5B-E repeated with RAW cells and/or Figure 5F-G repeated with RSL3 and LPS/Nigericin to further substantiate the claim.

4. Surface expression of WT vs K45Q and D53A NINJ1 is shown in Fig 2D and EV3 by western blot and microscopy. Flow cytometry would be more convincing to confirm equal surface expression.

5. The text describing the data in Figure 6A should refer to upper, middle, lower panels to ease the reading. It would also be better to replace "treated" with (CuOOH) like in the Figure headings, or "ferroptotic".

6. Minor: The title of Figure EV3 does not reflect what is shown in the Figure (mostly expression data)

7. Minor: Figure 5E: replace y-axis "Dextran count" with "3 kDa Dextran count" (to avoid having to read the legend when looking at the Figure)

Referee #3:

The current study delves into the functional role of ninjurin-1 (NINJ1) in the intricate process of ferroptosis, a regulated form of necrotic cell demise triggered by the iron-dependent buildup of oxidized phospholipids within cellular membranes. The findings of this investigation underscore the indispensability of NINJ1 in the initial loss of plasma membrane integrity, a pivotal event preceding the ultimate rupture of the plasma membrane (PMR). NINJ1 emerges as a critical mediator facilitating the liberation of cytosolic proteins and danger-associated molecular patterns (DAMPs) from cells undergoing ferroptosis, thus proposing that targeted modulation of NINJ1 could hold therapeutic promise in mitigating the inflammation associated with ferroptosis.

These noteworthy results encompass broad scientific significance and bear substantial therapeutic implications. However, a few salient observations warrant attention, potentially enhancing the clarity and robustness of the conclusions.

- In Figure 2-A and D, certain inconsistencies come to light. Notably, in Figure 2-A, the discernible distinction between WT and NINJ1-deficient cells after a 5-hour treatment with CuOOH contrasts with the less pronounced differences evident in Figure 2-D following a 2-hour treatment, where the presence of substantial standard deviations complicates interpretation. It is imperative that the experiment in Figure 2-D be replicated under conditions parallel to those in Figure 2-A, to definitively assert that reintroducing NINJ1 indeed averts PMR. Furthermore, duplicating these experiments utilizing RSL3 would provide informative corroboration.

- Moreover, it would be prudent to eliminate the potential influence of Ferrostatin-1 on NINJ1 oligomerization. A straightforward experiment could involve assessing whether Ferrostatin-1 impedes oligomerization in cells exposed to nigericin. Additionally, employing an alternative ferroptosis inhibitor with a distinct chemical structure could serve to underscore the robustness of the findings.

- The reliance on multiple assays to gauge the release of lactate dehydrogenase (LDH) as an indicator of PMR necessitates thorough consideration. Comparing the LDH protein levels of NINJ1-deficient cells against their WT counterparts might illuminate any differences, adding valuable depth to the study's assertions.

- Validating the findings within an established model of ferroptosis, such as HT1080 cells treated with RSL3, would notably fortify the study's impact within the research community.

Minor aspects:

To improve clarity, the legends accompanying the figures should explicitly detail the concentrations of the inducers and inhibitors used.

Finally, a thorough proofreading pass is advised to rectify redundant and ambiguous sentences. As an example, the sentence "indicating that in ferroptotic cells NINJ1 acts downstream of lipid peroxidation in ferroptotic cells" should be refined to avoid repetition and enhance clarity.

We would like to thank all three referees for highlighting the importance of our study, and for their insightful comments and suggestions. Below we provide a detailed answer to the individual requests.

Referee #1:

Ramos and colleagues study investigate the role of NINJ1 in the lysis of cells during ferroptosis. Using BMDMs as their main model, they demonstrate a role for NINJ1 in regulated cell lysis following two different ferroptotic stimuli, in line, NINJ1 is found to oligomerize. As a functional output of this, the authors demonstrate reduction of DAMP release from NINJ1 deficient BMDMs, leading them to speculate that NINJ1 regulates the inflammatory potential of ferroptosis. Overall, the authors present an interesting, timely study and the data largely support their conclusions. Nonetheless, a couple of points relating to the generality of their findings and the inflammatory consequences of NINJ1 expression during ferroptosis.

We thank the referee for the positive feedback on our manuscript, in particular for highlighting the importance and timeliness of our study. Below we explain how we addressed the points that were raised.

- The authors have primarily used BMDMs in the present study to study the role on NINJ1 in plasma membrane rupture thus a general requirement for NINJ1 in ferroptosis induced plasma membrane rupture is unclear. To address this, I would suggest investigating additional cell types (generating NINJ1 deficient cells via CRISPR/Cas9), measuring impact on ferroptosis induced cell lysis by LDH release.

We agree with the referee that it is important to show that the function of NINJ1 as executor of PMR is conserved in other cell types, besides macrophages. In our original submission we functionally analyzed the importance of NINJ1 in BMDMs, immortalized BMDMs (iBMDMs) and RAW264.7 cells and monitored NINJ1 clustering and function in HeLa cells.

In response to the referee's request, we have now expanded this analysis to additional cell types (namely MEFs and NIH/3T3 cells) and were able to demonstrate that following ferroptosis induction, NINJ1 mediates plasma membrane rupture (PMR) in these cell types as well.

Please see figures: **Fig. 3B, D, F and Appendix Figure 1A-C** of the revised manuscript.

With respect to the above point the authors demonstrate that plasma membrane lysis induced by CuOOH-treatment (but not RSL3) is NINJ1 dependent (suggestive of cell type and stimulus specificity), it is important to determine whether CuOOH is causing ferroptosis in these cells (i.e. can ferrostatin suppress).

We assume that the referee refers to the response of RAW264.7 cells, that showed a NINJ1 dependent response to CuOOH, but not to RSL3. As this indeed indicates cell type and stimulus-dependent specificity, we have explored this point by testing additional cell types (as mentioned above) and also by including ML162, a third ferroptosis activator.

Overall, this analysis confirmed that CuOOH, RSL3 and ML162 cause ferroptosis in the different cells types we had tested (BMDMs, RAW cells, MEFs and NIH/3T3), as cell death could be blocked with Fer-1 in each case. It also confirmed that cell lysis in BMDMs, MEFs and NIH/3T3 cells depends on NINJ1. On the other hand, RAW cells show only a mild dependence on NINJ1 when treated with CuOOH or ML162, and no NINJ1 dependency for RSL3 (which is in agreement with a previous report, PMID: 36898371).

What determines these cell-type specific differences remains unclear, but we speculate that there might be other factors that can permeabilize or rupture the plasma membrane similar to NINJ1 or that lipid peroxidation directly leads to PMR in certain cell types.

See figures: **Fig. 2C, Fig. 3B-F and Appendix Figure 1A-D** of the revised manuscript.

- The authors propose that NINJ1 regulates the inflammatory potential of ferroptosis (via promoting DAMP release), based on differential release of DAMPs from dying cells, though this is never formally test. This hypothesis could be directly investigated by transferring supt. from NINJ1 proficient/deficient cells triggered to undergo ferroptosis to recipient BMDMs and monitoring impact of inflammatory signalling pathways/transcriptional outputs.

We have addressed the role of NINJ1-dependent DAMP release by transferring supernatants from NINJ1 proficient/deficient cells on receiver cells and measuring induction of proinflammatory signaling as suggested by the referee. Our results (**Appendix Figure 4C,D**) show that the supernatants of WT cells do upregulate TNF α and IL-1 β expression more strongly than supernatants of NINJ1-KO cells. As the level of TNF α and IL-1 β induction at the chosen timepoint remains modest, a more thorough analysis of gene expression by RNAseq would more fully test the pro-inflammatory potential of ferroptosis. We believe such an RNA-seq experiment would be best suited for a follow-up study. Nevertheless, this experiment supports the notion that NINJ1-KO cells would cause less inflammation.

Referee #2:

In this study, Ramos et al. show evidence that the executor of plasma membrane rupture (PMR), NINJ1, is activated during ferroptosis and mediates the initial loss of membrane integrity, which is different from e.g., pyroptosis where gasdermin pores precedes NINJ1 activation and PMR. The significance of NINJ1 PMR in ferroptosis is suggested from it mediating the release of DAMPs which can drive tissue inflammation. The data are convincing, and the manuscript is well written and easy to read/follow. Some specific concerns/comments that should be addressed:

We would like to thank the referee for the positive feedback and the suggestions, which we address in detail below.

1. Two different inducers of ferroptosis are used: RSL3, an inhibitor of the antioxidant enzyme GPX4 which will inhibit the reduction of lipid peroxides (but probably incompletely), and Cumene hydroperoxide (CuOOH) which is a strong inducer of lipid peroxidation. From the results throughout the manuscript, it seems like CuOOH is a stronger inducer of ferroptosis than RSL3, and whereas Ferrostatin inhibits both inducers equally well, deletion of NINJ1 seems to more strongly impact CuOOH-induced ferroptosis (all steps). This is most prominent in RAW-cells (Figure EV2).

Could some of this be explained by suboptimal doses used for RSL3? How consistent is this in different cell types? Figure 5 shows differences in kinetics of PI influx, but it would be nice to see dose-responses of CuOOH and RSL3 in the cells used (BMDMs, RAWs and HeLa cells) with regards to cell death (Draq7 and LDH), and including Fer-1 and NINJ1-deficiency, to better understand these observations.

The referee raises an important point regarding the impact of NINJ1 on different ferroptosis inducers and in different cell types. To address this issue, we have done the following experiments:

- 1) We have expanded our analysis to a third ferroptosis inducer, ML162, which targets the same pathway as RSL3, and shown that it induces NINJ1-dependent lysis across several different cell lines which was always blocked by Fer-1. Please see figures: **Fig. 2C and Fig. 3B, D, F** of the revised manuscript.
- 2) We have titrated the three inducers, CuOOH, ML162 and RSL3 +/- Fer-1, and provide the PI uptake curves and LDH data in the revised manuscript. Please see figures: **Fig. 2C, Fig. 3B, D, F and Appendix Figure 1A-D (LDH release) and Fig. 5B-D, EV4A, C and Appendix Fig. 2A, B (PI analysis)** of the revised manuscript.
- 3) We have conducted this titration in several different cell lines, BMDMs, NIH/3T3 RAW 264.7 and MEF, and performed further experiments with the minimum concentration of activator required to specifically induce ferroptosis (blocked by Fer-1), to show that NINJ1 deficiency does not only impact LDH release and PI uptake in macrophages but also in other cell types.

2. Figure 4 shows reduced efficacy of Fer-1 in inhibiting lipid peroxidation starting 1h post ferroptosis induction (CuOOH), and at 3h there is no difference in Fer-1 treated and

untreated cells. Is this similar for RSL3? Is inhibition of cell death similarly reduced over time (effect of Fer-1) - or for other functions? E.g., in Figure 2C, Fer-1 still prevents DraG7 influx in response to CuOOH or RSL3 at 5h? How do the authors explain these data when, according to Figure 4, lipid peroxidation is at the same level in Fer-1 treated or untreated cells?

Indeed, the referee points out a conundrum, i.e. why did Fer-1 stop blocking lipid peroxidation at late timepoints in the FACS analysis, while it efficiently blocked cell death at the same timepoints in other experiments.

To address if this could be explained by a loss of activity in the batch of Fer-1 used for the BODIPY assay, we repeated the assay using CuOOH, RSL-3 and ML162 using fresh inhibitor. These repeats no longer showed a comparable loss of activity.

Furthermore, we realized that the high level of cell death with CuOOH and ML162 at 180 min post treatment results in an apparent drop in the percentage of BODIPY C11^{Ox} positive cells (see images below). The explanation for this is that the dying cells rupture and are no longer considered for the FACS analysis. Since this looks like cells lose BODIPY-C11^{Ox} staining, which is not correct, we have decided to only show FACS data for up to 120 min **Fig. 4B-D** of the revised manuscript.

3. Figure 5 is central for the claim that NINJ1 is required for initial membrane damage/permeability in ferroptosis. Given the differences seen in BMDMs and RAW cells, it would be nice to see Figure 5B-E repeated with RAW cells and/or Figure 5F-G repeated with RSL3 and LPS/Nigericin to further substantiate the claim.

As suggested, we have repeated the PI uptake in BMDMs and RAW cells (and NIH/3T3 and MEFs) with different ferroptosis inducers and at different concentrations: **Fig. EV4 and Appendix Figure 2** of the revised manuscript. The results support the conclusion that NINJ1 is required for initial membrane permeabilization, but also show that even in NINJ1-KO cells the membrane permeabilizes eventually by another mechanism, potentially due to microlesions caused by lipid peroxidation.

4. Surface expression of WT vs K45Q and D53A NINJ1 is shown in Fig 2D and EV3 by western blot and microscopy. Flow cytometry would be more convincing to confirm equal surface expression.

Unfortunately, the amount of anti-NINJ1 antibody that we received from Genetech Inc (gift of V. Dixit and N. Kayagaki) are very limited, and no commercial anti-NINJ1 antibody that works is available. Thus, we could not do the FACS analysis as suggested. However, to further substantiate that WT and K45Q and D53A NINJ1 localize to the plasma membrane, we show additional overview images of cells stained for NINJ1 (**Fig. EV2D** of the revised manuscript).

5. The text describing the data in Figure 6A should refer to upper, middle, lower panels to ease the reading. It would also be better to replace "treated" with (CuOOH) like in the Figure headings, or "ferroptotic".

Thank you for the suggestion, we have changed it accordingly.

6. Minor: The title of Figure EV3 does not reflect what is shown in the Figure (mostly expression data).

We have changed the title of this figure.

7. Minor: Figure 5E: replace y-axis "Dextran count" with "3 kDa Dextran count" (to avoid having to read the legend when looking at the Figure).

Thank you for the suggestion, we have changed it accordingly.

Referee #3:

The current study delves into the functional role of ninjurin-1 (NINJ1) in the intricate process of ferroptosis, a regulated form of necrotic cell demise triggered by the iron-dependent buildup of oxidized phospholipids within cellular membranes. The findings of this investigation underscore the indispensability of NINJ1 in the initial loss of plasma membrane integrity, a pivotal event preceding the ultimate rupture of the plasma membrane (PMR). NINJ1 emerges as a critical mediator facilitating the liberation of cytosolic proteins and danger-associated molecular patterns (DAMPs) from cells undergoing ferroptosis, thus proposing that targeted modulation of NINJ1 could hold therapeutic promise in mitigating the inflammation associated with ferroptosis. These noteworthy results encompass broad scientific significance and bear substantial therapeutic implications. However, a few salient observations warrant attention, potentially enhancing the clarity and robustness of the conclusions.

We thank the referee for the positive feedback on our manuscript and for highlighting the scientific importance and potential therapeutic implications of our study. Below we have addressed the points that were raised.

- In Figure 2-A and D, certain inconsistencies come to light. Notably, in Figure 2-A, the discernible distinction between WT and NINJ1-deficient cells after a 5-hour treatment with CuOOH contrasts with the less pronounced differences evident in Figure 2-D following a 2-hour treatment, where the presence of substantial standard deviations complicates interpretation. It is imperative that the experiment in Figure 2-D be replicated under conditions parallel to those in Figure 2-A, to definitively assert that reintroducing NINJ1 indeed averts PMR. Furthermore, duplicating these experiments utilizing RSL3 would provide informative corroboration.

The transduction of primary NINJ1 KO macrophages with NINJ1 expression vectors is a difficult experiment as we need to treat the cells 2x with virus during the differentiation into macrophages. Even then we manage to transduce only 30-40% of the cells (based on GFP expression levels – the plasmid contains an IRES GFP). It is thus not possible to reach the exact levels of LDH from complemented NINJ1 KO as we normally get from WT BMDMs, even after 5h of treatment.

We do agree though that it is important to be able to compare the LDH levels and treatment duration between **Fig. 2A and 2D**, and thus we provide the LDH levels from WT and NINJ1-KO BMDMs treated with CuOOH for the 2h timepoint in **Fig. 2F** of the revised manuscript.

Concerning RSL-3 treatments: As RSL3 is a much poorer inducer of ferroptosis in BMDMs, we would expect even lower levels of ferroptosis induction in NINJ1-reconstituted NINJ1 KO BMDMs. This, combined with higher variability that we observe in reconstituted BMDMs, would make the analysis difficult. Thus, we have not repeated the experiment with RSL-3.

- Moreover, it would be prudent to eliminate the potential influence of Ferrostatin-1 on NINJ1 oligomerization. A straightforward experiment could involve assessing whether Ferrostatin-1 impedes oligomerization in cells exposed to nigericin. Additionally, employing an alternative ferroptosis inhibitor with a distinct chemical structure could serve to underscore the robustness of the findings.

This is a valid concern and we have addressed it as suggested by the referee. Our results, included as **Fig. 1D and EV1B** in the revised manuscript, show that Fer-1 does neither impact NINJ1 clustering and oligomerization nor LDH release upon inflammasome activation with Nigericin.

As suggested, we have also treated WT BMDMs with Liproxstatin-1, another ferroptosis inhibitor, (**Fig. EV1A** of the revised manuscript) and confirmed reduction in LDH release for all the activators used in the paper.

- The reliance on multiple assays to gauge the release of lactate dehydrogenase (LDH) as an indicator of PMR necessitates thorough consideration. Comparing the LDH protein levels of NINJ1-deficient cells against their WT counterparts might illuminate any differences, adding valuable depth to the study's assertions.

As suggested by the referee, we have tested if basal LDH levels in WT or NINJ1-KO BMDMs differ. As shown in **Fig. EV2A** of the revised manuscript, we do not see any difference in the overall LDH levels between the two genotypes.

Thus, we can confirm that the differences in LDH release, we observe between WT or NINJ1 KO BMDMs are not caused by a loss of LDH expression in NINJ1-KO cells.

- Validating the findings within an established model of ferroptosis, such as HT1080 cells treated with RSL3, would notably fortify the study's impact within the research community.

As our initial submission used mostly macrophages (which are not that commonly used to study ferroptosis induction) to assess the role of NINJ1, we have analyzed additional cell types that have been used by others for ferroptosis studies previously (MEFs and NIH/3T3). These results confirm that NINJ1 is a driver of ferroptosis lysis in other cell types as well. Please see figures: **Fig. 3B, D, F and Appendix Fig. 1A-C** of the revised manuscript.

We have also tried to delete NINJ1 in HT1080 cells, but failed in our attempts (see below, panel A). To at least partially address the referee's comments, we have overexpressed hNINJ1 in HT1080 cells, and observed an accelerated plasma membrane permeabilization (pi uptake) after ferroptosis induction (see below, panels B-C). This indicates that NINJ1 can be activated by ferroptosis in HT1080 and that it can induce the loss of membrane integrity.

Minor aspects:

To improve clarity, the legends accompanying the figures should explicitly detail the concentrations of the inducers and inhibitors used.

Thank you for the suggestion, we have changed it accordingly.

Finally, a thorough proofreading pass is advised to rectify redundant and ambiguous sentences. As an example, the sentence "indicating that in ferroptotic cells NINJ1 acts downstream of lipid peroxidation in ferroptotic cells" should be refined to avoid repetition and enhance clarity.

Thank you for the suggestion, we have revised the manuscript accordingly.

Dear Prof. Broz,

Thank you for the submission of your revised manuscript to The EMBO Journal. We have now received the comments of the three referees that were asked to re-assess your study (included below). As you will see, all referees are satisfied with the revision, acknowledge that the previous concerns have been addressed satisfactorily and the study has been significantly improved, and they now support publication of the study.

From the editorial side, there are a few corrections and changes that we need from you before we can proceed with acceptance of the manuscript:

- Please make sure that all deposited data will be publicly available upon publication. You can now remove the referee access username and password from your Data Availability statement.
- Your Source Data files need to be re-organized to one zipped file/folder per main Figure. For EV and Appendix Figures, please ZIP together all source data.
- We noticed that there are some blank pages in the .docx file of your manuscript. Please check and correct if necessary.
- Please enter all relevant funding information in our online manuscript handling system. It should match exactly the information provided in the Acknowledgements section of your manuscript. Currently, the following information is missing from our online system: ERC-CoG 770988 (InflamCellDeath); (310030B_198005, 310030B_192523); ALTF 27-2022 to E.H. and ALTF 566-2022.
- We request authors to declare both actual and perceived competing interests in a competing interests statement (please review our policy here: <https://www.embopress.org/page/journal/14602075/authorguide#conflictsofinterest>). The heading of this statement should be "Disclosure and competing interests statement", and it should be placed after the "Materials and Methods" section.
- As per our journal's policy, "data not shown" (stated twice on pages 26 and 29 of your manuscript) is not permitted. All data referred to in the paper should be displayed in the main or Expanded View figures, or in the Appendix. Please add these data or change the text accordingly if these data are not central to the study and its conclusions.
- Please note that only the sections of the manuscript where the relevant information can be found should be provided in the last column of the Author Checklist. Please make sure that all information is included in the manuscript itself, and correct your checklist.
- Your Table EV1 should be renamed "Dataset EV1" and the corresponding callouts should be updated accordingly. Its legend should be removed from the manuscript file and only kept as a separate tab in the Excel file.
- The Appendix file should not be in a zip folder, but instead it should be uploaded as a single PDF file with the figures and their legends included in the file. The nomenclature should be Appendix Figure S1-S4, and the corresponding callouts should be updated accordingly throughout the manuscript. A brief Table of Contents with the correct page numbers should be provided on the first page of the Appendix.
- Please note that EMBO press papers are accompanied online by a synopsis image that is exactly 550 pixels wide and 300-600 pixels high (the height is variable). You can either show a model or key data in the synopsis image. The text needs to be readable at the final size. Please include this synopsis image in your re-submission.
- Please note that the figure legends are not ordered in a sequential manner for figures 3a-e (legends for figure panels 3c and 3e are provided before 3b and 3d, respectively). This needs to be rectified.
- Please note that the figure legends are not ordered in a sequential manner for figures EV 4a-c (legend for figure panel EV 4b is provided after 4c). This needs to be rectified.
- The movie files should be renamed Movie EV1-EV2, and the corresponding callouts should be updated accordingly. Their legends should be zipped together with each movie file.
- The manuscript sections are in the wrong order. Please follow the order: Title page, Abstract, Keywords, Introduction, Results, Discussion, Materials and Methods, Data availability, Acknowledgements, Disclosure and competing interests statement, References, Figure legends, Expanded View Figure legends.

As soon as these issues are resolved, I will contact you again to discuss with you a few suggestions for minor textual improvements in the title, abstract and synopsis text.

Please also note that as part of the EMBO publications' Transparent Editorial Process, The EMBO Journal publishes online a Peer Review File along with each accepted manuscript. This File will be published in conjunction with your paper and will include the referee reports, your point-by-point response and all pertinent correspondence relating to the manuscript. You can opt out of this by letting the editorial office know (contact@embojournal.org). If you do opt out, the Peer Review File link will point to the following statement: "No Peer Review File is available with this article, as the authors have chosen not to make the review process public in this case."

We look forward to seeing a final version of your manuscript as soon as possible. Please use this link to submit your revision: <https://emboj.msubmit.net/cgi-bin/main.plex>

Best regards,

Ioannis

Referee #1:

Authors have comprehensively addressed all the points I raised.

Referee #2:

The authors satisfactorily responded to the questions and concerns. The fact that they addressed all of the concerns with new experiments is highly appreciated, and has significantly improved the manuscript by substantiating their claims.

Referee #3:

The authors have satisfactorily addressed all my points. I have nothing else to add at this stage. Congratulations.

All editorial and formatting issues were resolved by the authors.

Dear Petr,

I am pleased to inform you that your manuscript has been accepted for publication in The EMBO Journal.

Best regards,

Ioannis
